# The online metacognitive control of decisions
Juliette Bénon [1], Douglas Lee [2], William Hopper[1], Morgan Verdeil[1], Mathias Pessiglione[1], Fabien Vinckier[1], Sebastien Bouret[1], Marion Rouault[1], Raphael Lebouc [1], Giovanni Pezzulo [3], Christiane Schreiweis[1], Eric Burguière [1] & Jean Daunizeau [1] ✉

Difficult decisions typically involve mental effort, which scales with the deployment of cognitive (e.g., mnesic, attentional) resources engaged in processing decision-relevant information. But how does the brain regulate mental effort? A possibility is that the brain optimizes a resource allocation problem, whereby the amount of invested resources balances its expected cost (i.e. effort) and benefit. Our working assumption is that subjective decision confidence serves as the benefit term of the resource allocation problem, hence the "metacognitive" nature of decision control. Here, we present a computational model for the *online metacognitive control of decisions* or oMCD. Formally, oMCD is a Markov Decision Process that optimally solves the ensuing resource allocation problem under agnostic assumptions about the inner workings of the underlying decision system. We demonstrate how this makes oMCD a quasi-optimal control policy for a broad class of decision processes, including -but not limited to- *progressive attribute integration*. We disclose oMCD's main properties (in terms of choice, confidence and response time), and show that they reproduce most established empirical results in the field of value-based decision making. Finally, we discuss the possible connections between oMCD and most prominent neurocognitive theories about decision control and mental effort regulation.

There is no such thing as a free lunch: obtaining reward typically requires investing effort. This holds even for mental tasks, which may involve mental effort for achieving success (in terms of, e.g., mnesic or attentional performance). Nevertheless, we sometimes invest very little mental effort, eventually rushing decisions and falling for all sorts of cognitive biases[1]. So how does the brain regulate mental effort? Recent theoretical neuroscience work proposes to view mental effort regulation as a resource allocation problem: namely, identifying the amount of cognitive resources that optimizes a cost/benefit tradeoff[2–4]. In this context, mental effort signals the subjective cost of investing resources, the aversiveness of which is balanced by the anticipated benefit. In conjunction with simple optimality principles, this idea has proven fruitful for understanding the relationship between mental effort and peoples' performance in various cognitive tasks, in particular those that involve cognitive control[5,6]. Recently, it was adapted to the specific case of value-based decision making, and framed as a self-contained computational model: the Metacognitive Control of Decisions or MCD[7].

The working assumption here is that decision confidence serves as the main benefit term of the resource allocation problem[8,9], hence the

"metacognitive" nature of decision control. On the one hand, this formalizes the regulating role of confidence in decision making, which has recently been empirically demonstrated in the context of perceptual evidence accumulation[10,11]. On the other hand, this apparently contrasts with standard treatments of value-based decision making, which insists on equating the benefit of value-based decisions with the value of the chosen option[12–14]. This notion is a priori appealing, because the purpose of investing resources into decisions is reducible to approaching reward and/or avoiding losses/punishments. Nevertheless, the benefit of such resource investments may be detached from the subjective evaluation of alternative options[15]. This is partly because the brain attaches subjective value to acquiring information about future rewards. In fact, this holds even when this information cannot be used to influence decision outcomes[16–18]. Recall that, in Marr's sense, any type of decision induces the same computational problem, i.e. the comparison of alternative options. In this view, *evidence-based* and *value-based* decisions simply differ w.r.t. to the underlying comparison criterion: the former relies on truthfulness judgments while the latter involves idiosyncratic preferences[19]. Hence, in both cases, the benefit of allocating resources

[1]Paris Brain Institute, Paris, France. [2]School of Electrical and Electronic Engineering, University College Dublin, Dublin, Ireland. [3]Institute of Cognitive Sciences and Technologies, Rome, Italy. ✉e-mail: jean.daunizeau@icm-institute.org

to decisions is to raise the chance of identifying the best option, i.e. confidence. In other words, if resource allocation aims at comparing alternative options, then decision confidence can be viewed as a probe for goal achievement. This is essentially a simplifying assumption, in the sense that it enables a unique computational architecture to control resource allocations, irrespective of the nature of the underlying decision-relevant computations.

In value-based decision making, confidence derives from the discriminability of uncertain value representations, which evolve over decision time as the brain processes more value-relevant information. Low confidence then induces a latent demand for mental effort: the brain refines uncertain value representations by deploying cognitive resources, until they reach an optimal confidence/effort trade-off. Interestingly, this mechanism was shown to explain the -otherwise surprising- phenomenon of choice-induced preference change[7]. More importantly, the MCD model makes quantitative out-of-sample predictions about many features of value-based decisions, including decision time, subjective feeling of effort, choice confidence and changes of mind. These predictions have already been tested -and validated- in a systematic manner, using a dedicated behavioral paradigm (Lee and Daunizeau, 2021). Despite its remarkable prediction accuracy, the original derivation of the model suffers from one main simplifying but limiting approximation: it assumes that MCD operates in a purely *prospective* manner, i.e., the MCD controller commits to a level of mental effort investment identified prior to the decision. In principle, this early commitment would follow from anticipating the prospective benefit (in terms of confidence gain) and cost of effort, given a prior or default representation of option values that would rely on fast/automatic/effortless processes[20]. The issue here, is twofold. First, it cannot explain variations in decision features (e.g., response time, choice confidence, etc.) that occur in the absence of changes in default preferences. Second, it is somehow sub-optimal, as it neglects *reactive* processes, which enable the MCD controller to re-evaluate – and improve on- the decision to stop or continue allocating resources, as new information is processed and value representations are updated. The current work addresses these limitations, effectively proposing an "online" variant of MCD which we coin oMCD.

As we will see, oMCD reduces to identifying the optimal policy for a specific instance of a known class of stochastic control problems: namely, "optimal stopping"[21]. This kind of problem can be solved using Markov Decision Processes or MDPs[22], under assumptions regarding the (stochastic) dynamics of costs and/or benefits. Although less concerned with the notion of mental effort, a similar MDP has already been derived for a specific type of "ideal" value-based decisions[14,23,24]. The underlying assumption here is threefold: (i) the system that computes option values is progressively "denoising" -in a Bayesian manner- its input value signals, (ii) the system that monitors and controls the decision knows how the underlying value computation system works, and (iii) the net benefit of decisions (i.e. the benefit discounted by decision time) is the estimated reward rate. The ensuing MDP is very similar to so-called Drift-Diffusion decision models[25,26], whereby the decision stops whenever the current estimate of option value differences reaches a threshold. Interestingly, the authors show that the assumptions (i), (ii) and (iii) imply that the optimal threshold is a decreasing function of time. This is not innocuous, since this predicts that decision confidence necessarily decreases with decision time, which is not always verified empirically[27]. In retrospect, these assumptions may thus be deemed too restrictive. In this work, we intend to generalize this kind of approaches by relaxing these three assumptions.

In particular, we will consider that the decision control system (i.e. the system that decides when to stop deliberating) has only limited information regarding the inner workings of the system that computes option values. We will show how decision confidence can serve both as an efficient titration for the benefit of resource investments and as a shortcut summary statistic for (hidden) value computations. That is, we will show that confidence monitoring is sufficient to operate quasi-optimal decision control for a wide class of value-based decision processes. We demonstrate the generalizability of the ensuing oMCD policy on two distinct decision scenarios. In the above "*Bayesian value denoising*" case, it replicates existing MDPs and extends

their repertoire of confidence/RT relationships. We also consider the case of value computation by *progressive attribute integration*[28–33]. As we will see, the latter scenario cannot be reduced to the *Bayesian value denoising* case. This is because the main source of uncertainty in value representations derive (as is the case for, e.g., forward planning) from the arbitrary incompleteness of value computations. We demonstrate that, for both decision scenarios, oMCD's control policy provides a close approximation to the ideal control policy, which requires complete knowledge of the underlying value computations. We also identify testable properties of oMCD control policies under both types of value computations, and show that they are reminiscent of empirical value-based decisions.

## Methods
As we will see below, deriving an optimal reactive variant of MCD requires specific mathematical developments, which falls under the frame of Markov decision processes[22]. But before we describe the oMCD model, let us first recall the prospective variant of MCD[7].

Note on ethics (see data re-analysis in the Results section): This work complies with all relevant ethical regulations and received formal approval from the INSERM Ethics Committee (CEEI-IRB00003888, decision no 16–333). All participants gave informed consent.

### The prospective MCD model
Note: this section is a summary of the mathematical derivation of the MCD model, which has already been published[7].

Let $z$ be the amount of cognitive (e.g., executive, mnemonic, or attentional) resources that serve to process value-relevant information. Allocating these resources will be associated with both a benefit $B(z)$, and a cost $C(z)$. As we will see, both are increasing functions of $z$: $B(z)$ derives from the refinement of internal representations of subjective values of alternative options or actions that compose the choice set, and $C(z)$ quantifies how aversive engaging cognitive resources is (mental effort). In line with the framework of *expected value of control*[2,4], we assume that the brain chooses to allocate the amount of resources $\hat{z}$ that optimizes the following cost-benefit trade-off:

$$\hat{z} = \arg\max_z \mathrm{E}[B(z) - C(z)] \tag{1}$$

where the expectation accounts for the anticipated impact of allocating resources into decision deliberation (this will be clarified below). Here, the benefit term is simply given by $B(z) = R \times P_c(z)$, where $P_c(z)$ is choice confidence and its weight $R$ quantifies the importance of making a confident decision. As we will see, $P_c(z)$ plays a pivotal role in the model, in that it captures the efficacy of allocating resources for processing value-relevant information. So, how do we define choice confidence?

We assume that the subjective evaluation of alternative options in the choice set is uncertain. In other words, the internal representations of values of alternative options are probabilistic. Such a probabilistic representation of value can be understood in terms of, for example, an uncertain prediction regarding the to-be-experienced value of a given option. In what follows, the probabilistic representation of option value $V_i$ takes the form of Gaussian probability density functions $p(V_i) = N(\mu_i, \sigma_i)$, where $\mu_i$ and $\sigma_i$ are the mode and the variance of the probabilistic value representation, respectively (and $i$ indexes alternative options in the choice set). This allows us to define choice confidence $P_c$ as the probability that the (predicted) experienced value of the (to be) chosen item is higher than that of the (to be) unchosen item. When the choice set is composed of two alternatives, $P_c$ is given by:

$$P_c \approx s\left(\frac{\pi|\Delta\mu|}{\sqrt{3(\sigma_1 + \sigma_2)}}\right) \tag{2}$$

where $s(x) = 1/1 + e^{-x}$ is the standard sigmoid mapping, and we assume that the choice follows the sign of the preference $\Delta\mu = \mu_1 - \mu_2$. Equation

(2) simply derives from a moment-matching approximation to the Gaussian cumulative density function[34]. Note that Eq. (2) implicitly assumes that the option with the highest value estimate is chosen. This satisfies the same formal criteria as for choice confidence in the context of evidence-based decisions[35].

We assume that the brain valuation system may, in some contexts, automatically generate uncertain estimates of options' value[36,37], before cognitive effort is invested in decision making. In what follows, $\mu_i^0$ and $\sigma_i^0$ are the mode and variance of the ensuing prior value representations. They yield an initial confidence level $P_c^0$. Importantly, this prior or default preference neglects existing value-relevant information that would require cognitive effort to be retrieved and processed[20].

Now, how can a decision control system anticipate the benefit of allocating resources to the decision process without knowing the details of the underlying value computations? Recall that the purpose of allocating resources is to process (yet unavailable) value-relevant information. The critical issue is thus to predict how both the uncertainty $\sigma_i$ and the modes $\mu_i$ of value representations will eventually change, before having actually allocated the resources (i.e., without having processed the information). In brief, allocating resources essentially has two impacts: (i) it decreases the uncertainty $\sigma_i$, and (ii) it perturbs the modes $\mu_i$ in a stochastic manner.

The former impact (i) derives from assuming that the amount of information that will be processed increases with the amount of allocated resources. This implies that the precision $1/\sigma_i(z)$ of a given probabilistic value representation necessarily increases with the amount of allocated resources, i.e.:

$$1/\sigma_i(z) = 1/\sigma_i^0 + \beta z \qquad (3)$$

where $1/\sigma_i^0$ is the prior precision of the representation (before any effort has been allocated), and $\beta$ controls the efficacy with which resources increase the precision of the value representation. More precisely, $\beta$ is the precision increase that follows from allocating a unitary amount of resources $z$. In what follows, we will refer to $\beta$ as "type #1 effort efficacy". Note that if $\beta = 0$, then mental effort brings no improvement in the precision of value representations.

The latter impact (ii) follows from acknowledging the fact that the control system cannot know how processing more value-relevant information will affect its preference before having allocated the corresponding resources. Let $\delta_i$ be the change in the position of the mode of the $i^{th}$ value representation, having allocated an amount $z$ of resources. The direction of the mode's perturbation $\delta_i$ cannot be predicted because it is tied to the information that is yet to be processed. However, a tenable assumption is to consider that the magnitude of the perturbation increases with the amount of information that will be processed. This reduces to stating that the variance of $\delta_i$ increases with $z$, i.e.:

$$\begin{aligned} \mu_i(z) &= \mu_i^0 + \delta_i \\ \delta_i &\sim N(0, \gamma z) \end{aligned} \qquad (4)$$

where $\mu_i^0$ is the mode of the value representation before any effort has been allocated, and $\gamma$ controls the relationship between the amount of allocated resources and the variance of the perturbation term $\delta$. The higher $\gamma$, the greater the expected perturbation of the mode for a given amount of allocated resources. In what follows, we will refer to $\gamma$ as "type #2 effort efficacy". Note that Eq. 4 treats the impact of future information processing as some form of random perturbation on the mode of the prior value representation. Importantly, Eq. 4 is not specific to the type of value computations that eventually perturbs the value modes. Our justification for this assumption is twofold: it is simple, and it captures the idea that the MCD controller is agnostic about how the allocated resources will be used by the underlying valuation/decision system. We will see that, in spite of this, the MCD controller can still make quasi-optimal predictions regarding the expected benefit of allocating resources, under very different value computation schemes.

Now, predicting the net effect of resource investment onto choice confidence (from Eqs. (3) and (4)) is not entirely trivial. On the one hand, allocating effort will increase the precision of value representations, which mechanically increases choice confidence, all other things being equal. On the other hand, allocating effort can either increase or decrease the absolute difference $|\Delta\mu(z)|$ between the modes (and hence increase or decrease choice confidence). This depends upon the direction of the perturbation term $\delta$, which is a priori unknown. Having said this, it is possible to derive the *expected* absolute mode difference (as well as its variance) that would follow from allocating an amount $z$ of resources:

$$\begin{cases} E[|\Delta\mu(z)|] = 2\sqrt{\frac{\gamma z}{\pi}}\exp\left(-\frac{|\Delta\mu^0|^2}{4\gamma z}\right) + \Delta\mu^0\left(2 \times s\left(\frac{\pi\Delta\mu^0}{\sqrt{6\gamma z}}\right) - 1\right) \\ V[|\Delta\mu(z)|] = 2\gamma z + |\Delta\mu^0|^2 - E[|\Delta\mu(z)|]^2 \end{cases} \qquad (5)$$

where we have used the expression for the first-order moment of the so-called "folded normal distribution". Importantly, $E[|\Delta\mu(z)|]$ is always greater than $|\Delta\mu^0|$ and increases monotonically with $z$ - as is $V[|\Delta\mu(z)|]$. In other words, allocating resources is expected to increase the value difference, even though the impact of the perturbation term can go either way.

Equation 5 now enables us to derive the expected confidence level $\bar{P}_c(z) \triangleq E[P_c]$ that would result from allocating the amount of resource $z$:

$$\bar{P}_c(z) \approx s\left(\frac{\lambda E[|\Delta\mu(z)|]}{\sqrt{1 + \frac{1}{2}(\lambda^2 V[|\Delta\mu(z)|])^{\frac{3}{4}}}}\right) \qquad (6)$$

where $\lambda = 1/\sqrt{3(\sigma_1(z) + \sigma_2(z))}$. Of course, $\bar{P}_c(0) = P_c^0$, i.e., investing no resources yields no confidence gain. Moreover, the expected choice confidence $\bar{P}_c(z)$ always increase with $z$, irrespective of the efficacy parameters, as long as $\beta \neq 0$ or $\gamma \neq 0$. Equation 6 is important, because it quantifies the expected benefit of resource allocation, before having processed the ensuing value-relevant information.

To complete the cost-benefit model, we simply assume that the cost of allocating resources to the decision process increases monotonically with the amount of resources, i.e.:

$$C(z) = \alpha z^\nu \qquad (7)$$

where $\alpha$ determines the effort cost of allocating a unitary amount of resources $z$ (we refer to $\alpha$ as the "unitary effort cost"), and $\nu$ effectively controls the range of resource investments that result in noticeable cost variations (we refer to $\nu$ as the "cost power").

Finally, the MCD-optimal resource allocation $\hat{z}$ is identified by replacing Eqs. (5), (6) and (7) into Eq. (1). This can be done before any resource has been invested, hence the *prospective* nature of metacognitive control, here.

## Online MCD: optimal control policy

We now augment this model, by assuming that the MCD controller re-evaluates the decision to stop or continue allocating resources, as value representations are being updated and online confidence is changing. This makes the ensuing *oMCD* model a *reactive* extension of the above "purely prospective" MCD model, which relieves the system from the constraint of effort investment pre-commitment.

Let $t$ be the current time within a decision. For simplicity, we assume that there is a linear relationship between deliberation time and resource investment, i.e.: $z = \kappa t$, where $\kappa$ is the amount of resources that is spent per unit of time. We refer to $\kappa$ as "effort intensity". By convention, the maximal decision time $T$ (the so-called *temporal horizon*) corresponds to the exhaustion of all available resources. This implies that $T = 1/\kappa$ because we consider normalized resources amounts.

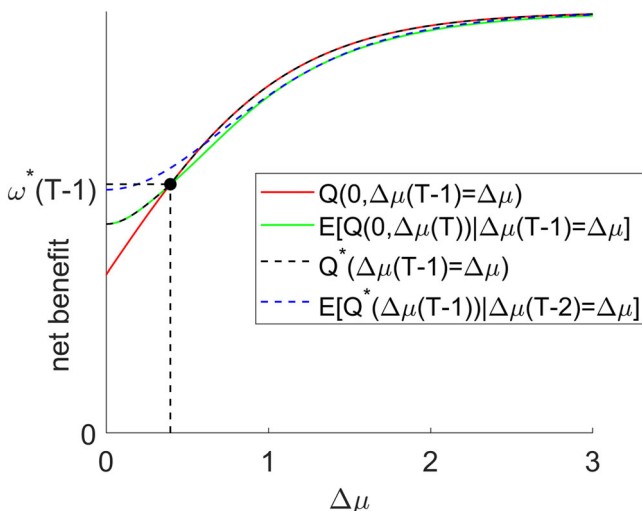

**Fig. 1 | Derivation of oMCD's optimal control policy.** Net benefits (y-axis) are plotted against the value mode difference (x-axis). The red and green lines show the net benefit if the system were stopping at $t = T - 1$, and the expected net benefit at $t = T - 1$. Finally, the dotted black line shows the optimal net benefit at $t = T - 1$, and the dotted blue line shows its expectation at $t = T - 2$ (see main text).

Now, at time $t$, the system holds probabilistic value representations with modes $\mu(t)$ and variance $\sigma(t)$. This yields the confidence level $P_c(\Delta\mu(t))$ given in Eq. (2) above, where we have made confidence an explicit function of $\Delta\mu(t)$ for mathematical convenience (see below).

This confidence level can be greater or smaller than the initial confidence level $P_c^0$, because new information regarding option values has been assimilated since the start of the deliberation. Of course, the system will anticipate that investing additional resources will increase its confidence (on average). But this may not always overcompensate the cost of spending more resources on the decision. Thus, how should the system determine whether to stop or to continue, in order to maximize the expected cost-benefit tradeoff? It turns out that this problem is one of *optimal stopping*, which is a special case of Markov Decision Processes[22,38]. As we will see, it can be solved recursively (backward in time) using Bellman's optimality principle[39].

Let $a(t) \in \{0, 1\}$ be the action that is taken at time $t$, where $a(t) = 0$ (resp. $a(t) = 1$) means that the system stops (resp. continues) deliberating. Let $Q(a(t), \Delta\mu(t))$ be the net benefit that the decision system would obtain at time $t$:

$$Q(a(t), \Delta\mu(t)) = \begin{cases} \underbrace{R \times P_c(\Delta\mu(t))}_{B(z)} - \underbrace{\alpha(\kappa t)^\nu}_{C(z)} & \text{if } a(t) = 0 \\ 0 & \text{otherwise} \end{cases} \quad (8)$$

where both the benefits $B(z)$ and costs $C(z)$ of resource investments have been rewritten in terms of decision time. Without loss of generality, Eq. (9) states that the net benefit of resource allocation is only realized when the system decides to stop ($a(t) = 0$). Note that $Q(a(t), \Delta\mu(t))$ is also a function of time (through the precision of value representations and effort cost), but we have ignored this dependency for the sake of notational conciseness.

A time $t$, the optimal control policy derives from a comparison between the net benefit of stopping now - i.e., $Q(0, \Delta\mu(t))$ - and some -yet undefined-threshold $\omega(t)$, which may depend upon time. Let $\pi_\omega(t)$ be the control policy (i.e., the temporal sequence of continue/stop decisions) that is induced by the threshold $\omega(t)$:

$$\pi_\omega(t) = \begin{cases} 0 \text{ if } Q(0, \Delta\mu(t)) \geq \omega(t) \\ 1 \quad \text{otherwise} \end{cases} \quad (9)$$

Finding the optimal control policy $\pi_\omega^*(t)$ thus reduces to finding the optimal threshold $\omega^*(t)$.

By definition, at $t = T$, the system stops deliberating irrespective of its current net benefit $Q(0, \Delta\mu(T))$. By convention, the optimal threshold $\omega^*(T)$ can thus be written as:

$$\begin{aligned} \omega^*(T) &= \min_{\Delta\mu(T)} Q(0, \Delta\mu(T)) \\ &= Q(0, 0, T) \\ &= R/2 - \alpha(\kappa T)^\nu \end{aligned} \quad (10)$$

Now, at $t = T - 1$, the net benefit $Q(0, \Delta\mu(T - 1))$ of stopping now can be compared to the expected net benefit $E[Q(0, \Delta\mu(T))|\Delta\mu(T - 1)]$ of stopping at time $t = T$, conditional on the current value mode difference $\Delta\mu(T - 1)$:

$$E[Q(0, \Delta\mu(T))|\Delta\mu(T - 1)] = R \times E[P_c(\Delta\mu(T))|\Delta\mu(T - 1)] - \alpha(\kappa T)^\nu \quad (11)$$

where the expectation is taken under the transition probability density $p(\Delta\mu(T)|\Delta\mu(T - 1))$ of the value mode difference for a unitary time increment ($\Delta t = 1 \iff \Delta z = \kappa$). This density derives from rewriting Eq. (4) in terms of the instantaneous change in the moments of the value representations. It is trivial to show that the corresponding first- and second-order moments are $E[\mu_i(t) - \mu_i(t - 1)] = 0$ and $E[(\mu_i(t) - \mu_i(t - 1))^2] = \gamma\kappa$, respectively. It follows that the transition probability density of the value mode difference is stationary (i.e. it does not depend upon time) and is given by:

$$p(\Delta\mu(t)|\Delta\mu(t - 1)) = N(\Delta\mu(t - 1), 2\gamma\kappa) \forall t > 1 \quad (12)$$

which is of course valid for $t = T$.

The optimal policy is to stop if $Q(0, \Delta\mu(T - 1)) \geq E[Q(0, \Delta\mu(T))|\Delta\mu(T - 1)]$, and to continue otherwise. Note that both $Q(0, \Delta\mu(T - 1))$ and $E[Q(0, \Delta\mu(T))|\Delta\mu(T - 1)]$ are deterministic functions of $\Delta\mu(T - 1)$. More precisely, they are both monotonically increasing with $\Delta\mu(T - 1)$ (see Fig. 1 below), because current confidence and expected future confidence monotonically increase with $\Delta\mu(T - 1)$. Critically, these functions have a different offset, i.e.: $Q(0, 0) < E[Q(0, \Delta\mu(T))|\Delta\mu(T - 1) = 0]$ as long as $\gamma > 0$. In addition, they eventually reach a different plateau, i.e.: $\lim_{\Delta\mu(T-1)\to\infty} Q(0, \Delta\mu(T - 1)) > \lim_{\Delta\mu(T-1)\to\infty} E[Q(0, \Delta\mu(T - 1))| \Delta\mu(T - 1)]$ as long as $\alpha > 0$. This is important, because this implies that there exists a critical value mode difference $\Delta\mu^*(T - 1)$ such that $Q(0, \Delta\mu^*(T - 1)) = E[Q(0, \Delta\mu(T))|\Delta\mu^*(T - 1)]$. The net benefit at that critical point is the optimal threshold at $t = T - 1$, i.e.: $\omega^*(T - 1) = Q(0, \Delta\mu^*(T - 1))$. This is exemplified in Fig. 1 below.

Now, let us move one step backward in time, at $t = T - 2$. Here again, the optimal policy is to stop if the current net benefit $Q(0, \Delta\mu(T - 2))$ is higher than the expected future net benefit $E[Q(a(T - 1), \Delta\mu(T - 1))|\Delta\mu(T - 2)]$, conditional on $\Delta\mu(T - 2)$. However, the latter now depends upon $a(T - 1)$, i.e., whether the system will later decide to stop or to continue:

$$\begin{aligned} &E[Q(a(T - 1), \Delta\mu(T - 1))|\Delta\mu(T - 2)] \\ &= \begin{cases} E[Q(0, \Delta\mu(T - 1))|\Delta\mu(T - 2)] \text{ if } a(T - 1) = 0 \\ E[E[Q(0, \Delta\mu(T))|\Delta\mu(T - 1)]|\Delta\mu(T - 2)] \text{ otherwise} \end{cases} \end{aligned} \quad (13)$$

The optimal control policy cannot be directly identified from Eq. (13). This is where we resort to Bellman's optimality principle: namely, whatever the current state and action are, the remaining actions of an optimal policy must also constitute an optimal policy with regard to the state resulting from the current action[39]. Practically speaking, the derivation of the optimal

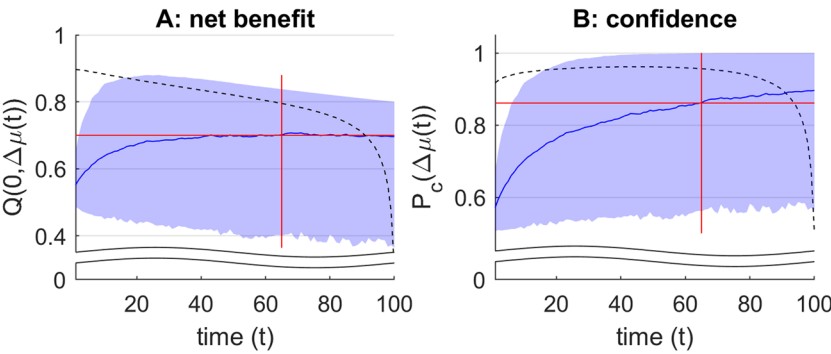

**Fig. 2 | oMCD's optimal control policy. A** The black dotted line shows the oMCD-optimal net benefit threshold. The blue line and shaded area depict the mean and standard deviation of net benefit dynamics (over the 1000 Monte-Carlo simulations), respectively. This reflects the possible variations of within-trial confidence dynamics. The vertical red line indicates the optimal resource allocation as obtained from the prospective variant of MCD, and the horizontal red line depicts the corresponding average net benefit level. **B** The black dotted line shows the oMCD-optimal confidence threshold. The blue line and shaded area depict the mean and standard deviation of decision confidence (over the same Monte-Carlo simulations), respectively. The horizontal red line depicts the average confidence level that corresponds to the optimal resource allocation under prospective MCD.

threshold at $t = T - 2$ is done under the constraint that oMCD's next action follows the optimal policy, i.e., $a(T - 1) = \pi_\omega^*(T - 1)$.

Let $Q^*(\Delta\mu(t)) \equiv Q(\pi_\omega^*(t), \Delta\mu(t))$ be the net benefit evaluated under the optimal policy at time $t$, which we refer to as the "optimal net benefit". Under Bellman's optimality principle, the optimal policy at $t = T - 2$ is to stop if the current net benefit $Q(0, \Delta\mu(T - 2))$ is higher than the expected optimal net benefit $E[Q^*(\Delta\mu(T - 1))|\Delta\mu(T - 2)]$, where the expectation is again taken under the transition probability density in Eq. (12).

Now, at time $t = T - 1$, the optimal net benefit is given by:

$$Q^*(\Delta\mu(T - 1)) \triangleq \max\{Q(0, \Delta\mu(T - 1)), E[Q(0, \Delta\mu(T))|\Delta\mu(T - 1)]\} \quad (14)$$

Note that $Q^*(\Delta\mu(T - 1))$ is just another function of $\Delta\mu(T - 1)$ (cf. dotted green curve in Fig. 1). This means that the only source of stochasticity in $Q^*(\Delta\mu(T - 1))$ comes from $\Delta\mu(T - 1)$, which can nonetheless be predicted (with some uncertainty), given the current value mode difference $\Delta\mu(T - 2)$. In turn, this makes the expected optimal net benefit $E[Q^*(\Delta\mu(T - 1))|\Delta\mu(T - 2)]$ a deterministic function of $\Delta\mu(T - 2)$. Again, as long as $\gamma > 0$ and $\alpha > 0$, there exists a critical value mode difference $\Delta\mu^*(T - 2)$ such that $Q(0, \Delta\mu^*(T - 2)) = E[Q^*(\Delta\mu(T - 1))|\Delta\mu^*(T - 2)]$. The net benefit at that critical point is the optimal threshold $\omega^*(T - 2)$ at $t = T - 2$.

In fact, the reasoning is the same for all times $t < T - 1$:

First, the expected optimal net benefit obeys the following backward recurrence relationship (Bellman equation for all $t < T - 1$):

$$E[Q^*(\Delta\mu(t))|\Delta\mu(t - 1)] = E[\max\{Q(0, \Delta\mu(t)), E[Q^*(\Delta\mu(t + 1))|\Delta\mu(t)]\}|\Delta\mu(t - 1)] \quad (15)$$

This equation is solved recursively backward in time, starting at the expected net benefit at $t = T - 1$, as given in Eq. (11). Both expectations in Eq. (15) are taken under the transition probability density $p(\Delta\mu(t)|\Delta\mu(t - 1))$ of the value mode difference under a unitary resource investment (cf. Equation (12)).

Second, the optimal threshold at time $t$ is given by:

$$\omega^*(t) = Q(0, \Delta\mu^*(t)) \quad (16)$$

where $\Delta\mu^*(t)$ is the critical value mode difference, i.e., $\Delta\mu^*(t)$ is such that:

$$Q(0, \Delta\mu^*(t)) = E[Q^*(\Delta\mu(t + 1))|\Delta\mu(t) = \Delta\mu^*(t)] \quad (17)$$

Since the net benefit is a deterministic function of decision confidence, the oMCD-optimal threshold $\omega^*(t)$ for net benefits can be transformed into an oMCD-optimal confidence threshold $\omega_P^*(t)$. Replacing the net benefit

with the optimal threshold $\omega^*(t)$ and confidence with $\omega_P^*(t)$ in Eq. 9 yields:

$$\omega_P^*(t) = \frac{\omega^*(t) + \alpha(\kappa t)^\nu}{R} \quad (18)$$

At any point in time, comparing the net benefit $Q(0, \Delta\mu(t))$ of resource allocation to $\omega^*(t)$ is exactly equivalent to comparing the current confidence level $P_c(t)$ to $\omega_P^*(t)$. In other terms, the optimal control policy (cf. Equation (10)) can be rewritten as:

$$\pi_\omega^*(t) = \begin{cases} 0 \text{ if } P_c(t) \geq \omega_P^*(t) \\ 1 \quad \text{otherwise} \end{cases} \quad (19)$$

This highlights the central role of confidence, whose monitoring (during deliberation) is a sufficient condition for operating optimal decision control. In turn, this greatly simplifies the decision control architecture because knowledge about the underlying decision-relevant computations is not required. As we will see later, oMCD is flexible (i.e. it encompasses many kinds of decision processes) and robust to deviations from its working assumptions (i.e. it provides a tight approximation to optimal control under alternative settings of the resource allocation problem).

This closes the derivation of oMCD's optimal control policy.

Although the derivation of oMCD's optimal control policy is agnostic w.r.t. the underlying value computations, it still requires some prior information regarding the upcoming information processing: namely, prior moments of value representations, type #1 and #2 effort efficacies, decision importance, unitary effort cost and cost power. This means that oMCD implicitly includes a *prospective* component, which is used to decide how to optimally *react* to a particular (stochastic) internal state of confidence. In other terms, one can think of oMCD as a mixed prospective/reactive policy, whose prospective component is the shape of the confidence threshold temporal dynamics.

Figure 2 below shows a representative instance of oMCD's optimal control policy, from 1000 Monte-Carlo simulations (using decision parameters R = 1, α = 0.2, β = 1, γ = 4, κ = 1/100, υ = 0.5, $\sigma_0$ = 1).

First, one can see that oMCD's optimal confidence threshold $\omega_P^*(t)$ lies above the average confidence level $\bar{P}_c(t)$ of its prospective variant (cf. Equation 6, whose Monte-Carlo estimate is depicted by the blue line in panel B). This means that oMCD's control policy would, in most cases, demand higher confidence than prospective MCD. Importantly however, oMCD's policy is sensitive to unpredictable fluctuations in the trajectory of value modes, which will induce variations in resource investments (or, equivalently, response times). This enables oMCD to exploit favorable variations in confidence if they eventually reach the threshold sooner than expected.

**Fig. 3 | Impact of decision parameters on oMCD's optimal confidence threshold dynamics. A** Effect of type #1 effort efficacy. Optimal confidence threshold (y-axis, black dots) is plotted against decision time (x-axis), for different β levels (color code). **B** Effect of type #2 effort efficacy, same format. **C** Effect of unitary effort cost, same format. **D** Effect of cost power, same format.

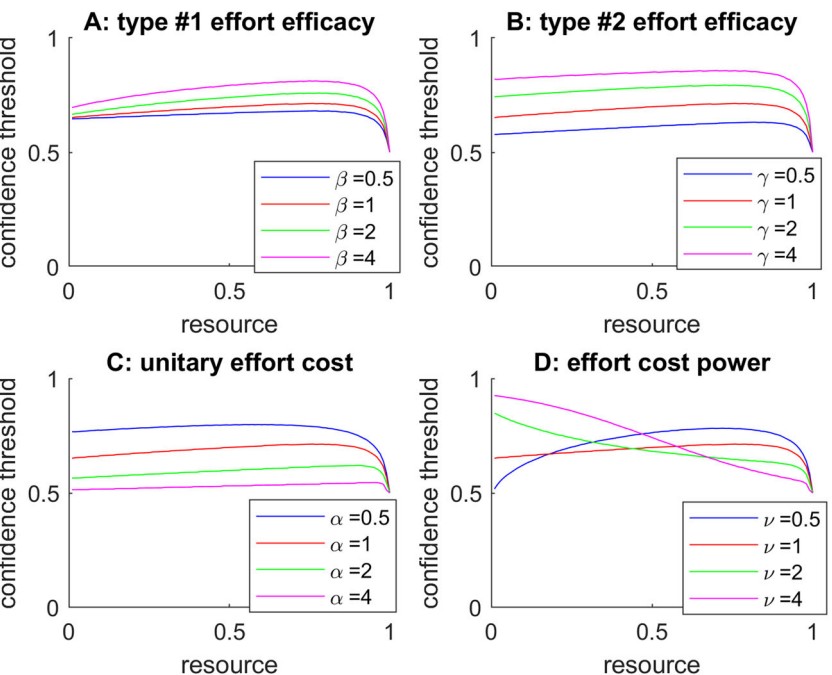

Note that the confidence threshold $\omega_p^*(t)$ is, by construction, the confidence level that the system achieves when committing to its decision. This means that, under oMCD's policy, the relationship between reported confidence levels and response times is entirely determined by the shape of the optimal threshold dynamics. In this example, this relationship will be mostly negative, i.e., reported confidence levels tend to decrease when response times increase. This is despite the fact that average confidence $\bar{P}_c(t)$ always increases as decision time unfolds, as long as effort efficacy parameters are nonzero. In other words, the overt relationship between response times and reported confidence levels (across trials) may be qualitatively different from the covert temporal dynamics of confidence during decision deliberation.

So what is the impact of decision parameter on oMCD's confidence threshold dynamics? This is summarized in Fig. 3 below, where we systematically vary each parameter in turn (when setting all the others to unity).

The net effect of increasing effort efficacy (either type #1 or type #2) is to increase the absolute confidence threshold. In other terms, the demand for confidence increases with effort efficacy. In contrast, the demand for confidence decreases with unitary effort cost. Note that the effect of increasing decision importance (not shown) is exactly the same as that of decreasing unitary effort cost. Importantly, the shape of the confidence threshold dynamics is approximately invariant to changes in effort efficacy or unitary effort cost.

The only parameter that eventually changes the qualitative dynamics of oMCD's optimal confidence threshold is the effort cost power (panel D). In brief, increasing the cost power tends to decrease the initial slope of oMCD's confidence threshold dynamics. Here, the latter eventually falls below zero (i.e., the confidence threshold decreases with decision time) when the effort cost becomes superlinear ($\nu > 1$). This is because, in this case, late resource investments are comparatively more costly than early ones.

Note that, in contrast to effort efficacies, effort cost parameters can be altered without changing the dynamics of expected confidence. In other terms, the shape of the relationship between decision time and confidence is, for the most part, independent from the inner workings of the underlying decision system.

Let us now relate the MCD framework to standard decision processes, which differ in terms of their respective value computations.

## How does MCD relate to standard decision processes?

By itself, the MCD framework does not commit to any specific assumption regarding how value-relevant information is processed. Nevertheless, the properties of decisions that are controlled through MCD actually depend upon how probabilistic value representations change over time. In what follows, we focus on two specific scenarios of value computations, and disclose their connection with MCD.

### Bayesian value denoising

Let us first consider the *Bayesian value denoising* case, in which value representations are updated Bayesian beliefs on a hidden value signal. Note that, in this case, the optimal control rule - for maximizing expected reward rate - reduces to a specific instance of so-called *drift-diffusion decision* models with decaying bounds on the estimated value difference[14,24].

Assume that, at each time point, the decision system receives an unreliable copy $y(t)$ of the (hidden) value $V$ of each alternative option. More precisely, $y(t)$ is a noisy input signal that is centered on $V$, i.e.:$y(t) = V + \varepsilon(t)$, where the random noise term $\varepsilon(t)$ is i.i.d. Gaussian with zero mean and variance $\Sigma$ (and we have dropped the option indexing for notational simplicity). One may think of $\Sigma$ as measuring the (lack of) reliability of the input value signal. This induces the following likelihood function for the hidden value: $p(y(t)|V) = N(V, \Sigma)$. Finally, assume that the decision system holds a Gaussian prior belief about the hidden options' value, i.e.: $p(V) = N(\mu_0, \sigma_0)$, where $\mu_0$ and $\sigma_0$ are the corresponding prior mean and variance. At time $t$, a Bayesian observer would assimilate the series of noisy signals to derive a probabilistic (posterior) representation $p(V|y(1), ..., y(t)) = N(\mu(t), \sigma(t))$ of hidden options' values with the following mean and variance[40]:

$$\begin{cases} \mu(t) = \mu_0 + \tilde{\delta}(t) \\ \sigma(t) = \dfrac{1}{\dfrac{1}{\sigma_0} + t \times \dfrac{1}{\Sigma}} \end{cases} \quad (20)$$

where the perturbation $\tilde{\delta}$ of the value mode is given by:

$$\tilde{\delta}(t) = \frac{1}{\frac{\Sigma}{\sigma_0} + t} \sum_{t'=1}^{t} (y(t) - \mu_0) \quad (21)$$

Equation 21 specifies what the perturbation to the value mode would be, if the underlying value computation was a process of *Bayesian value denoising*, whose outcome is the posterior estimate $\mu(t) = E[V|y(1), ..., y(t)]$ of value. In brief, Eq. (21) states that the value mode changes in proportion to prediction errors (i.e., $y(t) - \mu_0$), which the Bayesian observer accumulates while sampling more input value signals. The stochasticity of the value mode's perturbation $\tilde{\delta}$ is driven by the random noise term $\varepsilon$ in the incoming noisy value signal. Conditioned on the hidden value $V$, it is easy to show that $E[\tilde{\delta}|V] \propto V - \mu_0$. This implies that the random walk in Eq. (21) actually has a nonzero drift that is proportional to the hidden value. Importantly however, the Bayesian observer does not know what the hidden value $V$ is. Prior to observing noisy value signals, its expectation is simply that $E[y] = E[V] = \mu_0$ and therefore $E[\tilde{\delta}] = 0$. In fact, this holds true at any time $t$: the Bayesian observer's expectation about the future change in its value belief mode, i.e. $E[\mu(t+1) - \mu(t)|y(1), ..., y(t)]$, is always zero, because its expectation about the next value signal reduces to her current value mode, i.e. $E[y(t+1)|y(1), ..., y(t)] = \mu(t)$. In other words, although the modes' perturbation $\tilde{\delta}$ actually have a nonzero mean (as long as $V$ deviates from the mode of the observer's belief), the Bayesian observer's expectation about its future realizations is always zero.

Nevertheless, the Bayesian observer can accurately predict how the precision of its belief will change with time. Comparing Eqs. (3) and (20) suggests that, under the *Bayesian value denoising* scenario, type #1 effort efficacy reduces to: $\beta = 1/\kappa\Sigma$. This means that type #1 effort efficacy simply increases with the reliability of the input value signal.

In addition, although the Bayesian observer cannot anticipate in what direction the to-be-sampled signal $y(t)$ will modify the mode of its posterior belief, it can derive a prediction over the magnitude of the perturbation:

$$E[\tilde{\delta}(t)^2] = t \times \frac{\Sigma + t\sigma_0}{\left(\frac{\Sigma}{\sigma_0} + t\right)^2} \qquad (22)$$

where the expectation is derived under the agent's prior belief about the hidden value. Now, Eq. (4) defines type #2 effort efficacy in terms of the ratio $E[\tilde{\delta}(t)^2]/\kappa t$ of expected change magnitude over effort investment (where $z = \kappa t$). Note that, under Eq. (22), this quantity varies as a function of decision time. Thus, under the *Bayesian value denoising* scenario, type #2 effort efficacy can be approximated as its sample average over all admissible decision times, i.e.: $\gamma \approx 1/T\sum_{t=1}^{T}(\Sigma + t\sigma_0)/(\Sigma/\sigma_0 + t)^2\kappa$. This is only an approximation of course, since $E[\tilde{\delta}(t)^2]$ eventually tails off as time increases, because noisy value signals that are sampled later in time have a smaller effect on the posterior mode. In other words, were the MCD controller to know about the inner computations of the underlying value updating system, it would rely on Eq. (22) rather than on Eq. (4). The ensuing ideal control policy is summarized in the Supplementary Methods 1 in the Supplementary Information.

**The progressive attribute integration case**

Second, let us consider another type of value computation, which essentially proceeds from progressively integrating the value-relevant attributes of choice options. This typically happens when choice options can be decomposed into multiple dimensions that may conflict with each other (cf., e.g., tastiness versus healthiness for food items).

Let $x_1, ..., x_k$ be the set of $k$ such value-relevant attributes, the combination of which is specific to each option. Assume that the decision system constructs the value of alternative options according to a weighted sum of attributes, i.e.: $V = \sum_k w_k \times x_k$, where the attribute weights $w_k$ are the same for all options. Assume that each attribute is sampled from a Gaussian distribution with mean $\eta_k$ and variance $\varsigma_k$, i.e. $p(x_k) = N(\eta_k, \varsigma_k)$. Finally, assume that attributes are available to the decision system one at a time, i.e. decision time steps co-occur with attribute-disclosing events. For the sake of simplicity, we set the decision's temporal horizon to $T = k$, i.e. we focus on the decision to stop (potentially prematurely) the integration of all available value-relevant attributes. In what follows, we refer to this scenario as the *progressive attribute integration* model.

In the absence of default preferences, the system holds a prior representation about the options' value that is maximally uninformative. This is because, prior to any value computation, any combination of value-relevant attributes is admissible, and the system did not disclose the options' attributes yet. The first two moments of the system's prior value representation $p(V) = N(\mu_0, \sigma_0)$ are thus given by:

$$\begin{cases} \mu_0 = \sum_{k'=1}^{k} w_{k'} \times \eta_k \\ \sigma_0 = \sum_{k'=1}^{k} w_{k'}^2 \times \varsigma_k \end{cases} \qquad (23)$$

where of $k$ is the number of value-relevant attributes.

Now, as time unfolds and the decision system discloses the value-relevant attributes, it progressively removes sources of uncertainty about the value of alternative options. In principle, if the system reaches the temporal horizon, then it knows all the attributes and can evaluate the alternative options with infinite precision. However, as long as some attributes are missing, value representations remain uncertain. Let $K(t)$ be the set of attribute indices that have been available to the decision system up until time $t$. At time $t$, the decision system thus holds an updated probabilistic representation of value $p(V|x_{K(t)}) = N(\mu(t), \sigma(t))$ with the following mean and variance:

$$\begin{cases} \mu(t) = \mu_0 + \tilde{\delta}(t) \\ \sigma(t) = \sigma_0 - \sum_{k' \in K(t)} w_{k'}^2 \times \varsigma_{k'} \end{cases} \qquad (24)$$

where the change in the value mode is simply given by:

$$\tilde{\delta}(t) = \sum_{k' \in K(t)} w_{k'} \times (x_{k'} - \eta_{k'}) \qquad (25)$$

As before, Eq. 25 specifies what the perturbation to the value mode would be, if the underlying value computation was a process of *progressive attribution integration*, whose outcome is the value estimate $\mu(t)$. Note that here, variability in mode perturbations does not arise from some form of stochasticity or unreliability of input signals, as is the case for the *Bayesian value denoising* scenario above. Rather, it derives from the arbitrariness of the permutation order with which attributes become available for options' evaluation. However, should the full set of attributes eventually be disclosed, the estimated value would be $\mu(T) = \sum_{k'}^{k} w_{k'} \times x_{k'}$, with full certainty ($\sigma(T) = 0$).

Here again, the decision system cannot anticipate in which direction the future value mode will change, i.e. its expectation over future mode changes always is $E[\tilde{\delta}(t)] = 0$ at any point in time (because $E[x_k] = \eta_k$). Nevertheless, it can derive a prediction over the magnitude of the perturbation, by averaging over all possible permutation orders:

$$\begin{aligned} E[\tilde{\delta}(t)^2] &= \frac{t}{k} \sum_{k'=1}^{k} w_{k'}^2 \times \varsigma_{k'} \\ &= t\,\sigma_0 \end{aligned} \qquad (26)$$

Comparing Eqs. (4) and (26) suggests that, under the *progressive attribute integration* scenario, type #2 effort efficacy simplifies to: $\gamma = \sigma_0$. This means that type #2 effort efficacy simply scales with the expected range of attributes' variation. This also implies that, in contrast to the above *value denoising* case, the transition probability density of value modes under the *progressive attribute* integration scenario is stationary and complies with oMCD's assumption (cf. Equation (12)).

What about type #1 effort efficacy? Note that one cannot directly compare Eq. (24) to Eq. (4), because of the arbitrariness of the order of attribute-disclosing events. In fact, this arbitrariness implies that the dynamics of value variances is decreasing with time but stochastic. Although

**Fig. 4 | the performance of oMCD's optimal control policy. A** A violin plot of the distribution of resources invested (y-axis) is shown under oMCD (black), prospective MCD (red), or oracle (green) policies. Horizontal lines and shaded areas depict sample mean and standard deviation, respectively. **B** Average confidence level at the time of decision, same format. **C** The average net benefits, same format. **D** Mean achieved confidence (y-axis) is plotted against resource investment deciles (x-axis) for all control policies (oMCD: black, MCD: red, oracle: green). Errobars show standard deviation around the mean (s.e.m.). The black dotted line shows oMCD's optimal confidence threshold.

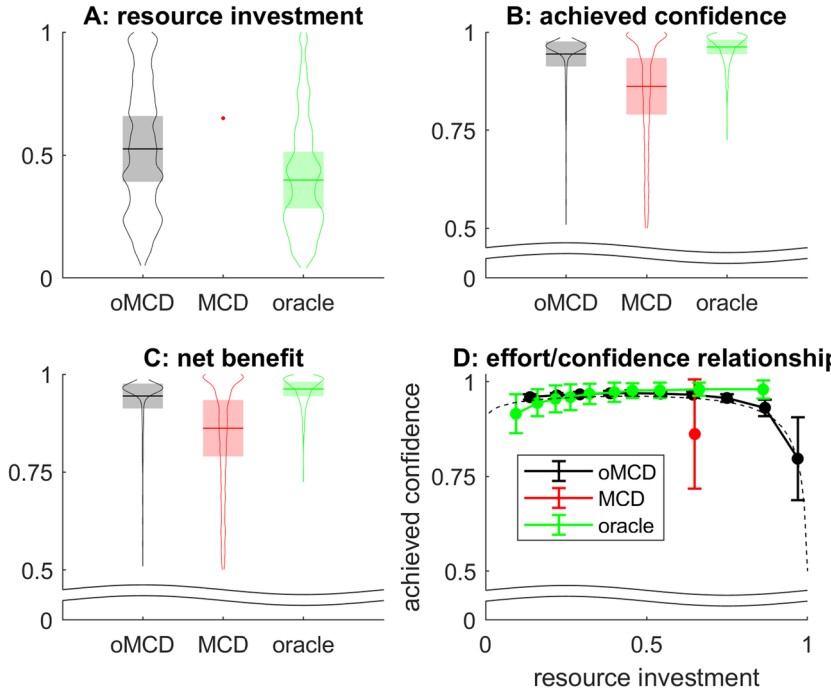

oMCD is neglecting this stochasticity, type #1 efficacy can be derived from the first-order moment of value variance dynamics. Accordingly, averaging over all possible permutations yields the following expected change in precision: $E[1/\sigma(t) - 1/\sigma_0] \approx t \times 1/\sigma_0(k - t)$. Using the same logic as above, this suggests that type #1 effort efficacy can now be approximated as: $\beta \approx 1/(k - 1)\sum_{t=1}^{k-1} 1/\kappa \sigma_0(k - t)$. Note that we have removed the time horizon from averaging over admissible decision times, since it induces a singularity (infinite precision).

Importantly, the *progressive attribute integration* scenario implies that both first- and second-order moments of value representations follow stochastic dynamics. This means that the ideal control policy does not reduce to a single threshold (on either net benefits or confidence), but rather unfolds onto the bidimensional space spanned by both moments of value representations. This makes the *progressive attribute* integration scenario qualitatively different from the *Bayesian value denoising* case. We refer the interested reader to the Supplementary Methods 2 in the Supplementary Information for details regarding the mathematical derivation of the ideal control policy under *progressive attribute integration*.

One can see that the definition of type #1 and type #2 effort efficacies depends upon the way in which the decision process perturbs the value representations (the above scenarios are just two examples out of many possible forms of value computations). In principle, optimal control would thus require variants of MCD controllers that are tailored to the underlying decision system. For the sake of completeness, the derivation of such ideal control policies are summarized in Appendices 1 and 2. In this context, the MCD architecture that we propose provides an efficient alternative, which generalizes across decision processes and still operates quasi-optimal decision control (see below). The only requirement here, is to calibrate the MCD controller over a few decision trials to learn effort efficacy parameters. Note that such calibration is expected to be very quick (at the limit: only one decision trial), because effort efficacies can be learned on within-trial dynamics (of value representations). This is effectively what we have done here, in an analytical manner, when deriving approximations for the effort efficacy parameters under distinct decision scenarios.

### Reporting summary

Further information on research design is available in the Nature Portfolio Reporting Summary linked to this article.

## Results

In the previous section of this manuscript, we derived the online, dual prospective/reactive variant of MCD (and disclosed its connection with two exemplar decision systems). We now wish to illustrate its properties.

### How do prospective MCD and oMCD differ?

Formally speaking, online/reactive and prospective MCD policies are solving the same resource allocation problem, i.e. they both aim at stopping resource investment when its net benefits are maximal. At this point, one may thus ask whether oMCD produces better decisions than prospective MCD, which operates by committing to a predefined resource investment. More precisely, under prospective MCD, the decision stops when the expected net benefit is maximal, which is evaluated at the onset of the decision (this corresponds to the red vertical line in Fig. 2). But does oMCD yield higher net benefits than prospective MCD (on average)?

To answer this question, we resort to Monte-Carlo simulations. In brief, we simulate a particular decision trial in terms of the stochastic dynamics of value representations, according to Eqs. (3) and (4), using the same decision parameters as for Fig. 2. At each time step, oMCD's policy proceeds by comparing the ensuing confidence level to the optimal confidence threshold. When the confidence threshold is reached, we store the resource investment, as well as the ensuing confidence level and net benefit. We proceed similarly for prospective MCD, except that resource investment is defined according to Eq. (1). We then repeat the procedure to evaluate the average confidence levels, amounts of invested resources, and net benefits induced by both MCD variants. These are summarized in Fig. 4 below, where the averages are taken over 500 sample path trajectories of value modes. Note: as a reference, we also compare MCD control policies to a so-called "oracle" dummy policy, which retrospectively identifies the net benefit apex, i.e. the time at which the stochastic trajectory of net benefits is maximal. This provides an upper (though unachievable) bound to the expected net benefit of any online control policy.

One can see that oMCD tends to invest fewer resources and yet achieves higher confidence than prospective MCD (on average). In turn, the ensuing average net benefit is lower for prospective MCD than for oMCD (which is closer to the oracle). Unsurprisingly, under oMCD, the statistical relationship between resource investments and reported confidence levels unfolds along the dynamics of the optimal confidence threshold. In this setting, decisions that take longer eventually yield lower confidence

**Fig. 5 | Comparison between prospective MCD and oMCD. A** The amount of resources invested under the prospective variant of MCD (x-axis) is plotted against the average amount of resources invested under oMCD (y-axis). Each dot corresponds to a specific set of decision parameters (200 samples). The color code indicates type #2 effort efficacy (blue: low $\gamma$, red: high $\gamma$). **B** Decision confidence, same format. **C** Net benefit, same format.

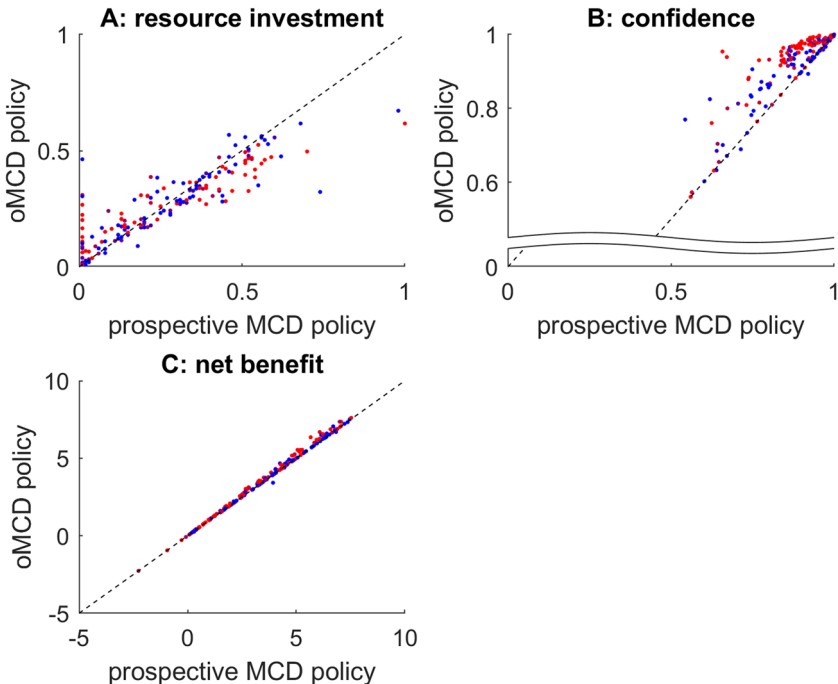

(although this actually depends upon decision parameters, see Fig. 3). For prospective MCD, there is no such relationship because resource investment is fixed once decision parameters are set.

So do these observations generalize over decision parameter settings? To answer this question, we repeat the same analysis as above, under 200 random settings of all decision parameters. Figure 5 below summarizes the results of this Monte-Carlo simulations series.

One can see that the impact of decision parameters on resource investment and confidence is very similar under both MCD variants. This is important, because this means that the known properties of prospective MCD[7] generalize to oMCD. In addition, oMCD's optimal control policy tends to yield lower resource investments and higher confidence levels than prospective MCD. Both effects almost compensate each other, but oMCD tends to provide a small but systematic improvement on the ensuing net benefit, which typically increases with type #2 effort efficacy ($\gamma$). This is because increasing $\gamma$ increases the stochasticity of value mode dynamics, which provides oMCD with more opportunities to exploit favorable variations in confidence (cf. panel B).

Now, when compared to prospective MCD, oMCD possesses a unique feature: the potentially nontrivial statistical relationship between decision confidence and resource investments (as proxied using, e.g., response times), *across trials with identical decision parameters*. This was already exemplified in Fig. 4 above (cf. panel D).

To make this distinction clearer, we performed another set of simulations aiming at evaluating the impact of decision difficulty. Note that difficult decisions can be defined as those decisions where the reliability of value representations improve very slowly. Within the MCD framework, increasing decision difficulty can thus be modelled by decreasing type #1 effort efficacy. We systematically varied $\beta$ from 2 to 8 (having set all the other decision parameters to 4), simulated 500 sample path trajectories of value mode dynamics for each difficulty level, and evaluated the ensuing effort investments and achieved confidence levels. Figure 6 below summarizes the simulation results.

One can see that the net effect of increasing decision difficulty (or equivalently, decreasing type #1 effort efficacy) is to increase resource investment and decrease confidence. This holds for both oMCD and its prospective variant. This means that, on average, reported confidence levels will tend to correlate negatively with resource investments, *across difficulty levels* (at least for this setting of decision parameters). However, for oMCD,

this negative relationship between resource investments and reported confidence levels is also true *within each difficulty level* (across trials). This has no equivalent under prospective MCD. In addition, the shape of this relationship is preserved across difficulty levels. This is because type #1 effort efficacy induces rather small distortions on oMCD's confidence thresholds (cf. Figure 3 above).

Figure 6 also reveals how oMCD's optimal control policy prospectively anticipates the impact of decision difficulty. In brief, the decay rate of oMCD's confidence threshold increases with decision difficulty, because expected confidence gains become more costly. However, this is over-compensated by the corresponding decrease in the ascent rate of expected confidence, which will delay the time at which confidence eventually reaches the optimal threshold. This eventually determines the way oMCD trades effort against confidence: difficult decisions are given more deliberation time than easy decisions (this is also true for prospective MCD).

Note that the effect of difficulty on resource investment, as well as the shape of the effort/confidence relationship, depends on the setting of decision parameters. In other words, these effects do not generalize to all decision parameter settings. For example, increasing decision difficulty will eventually decrease resource investments. Also, the sign of the correlation between confidence and resource investments across difficulty levels may not always align with the sign of this correlation within each difficulty level.

## How optimal is oMCD's policy?
One of oMCD's main claims is that it is possible to derive a quasi-optimal decision control policy, without detailed knowledge of the underlying value computations. But how well does oMCD perform, when compared to ideal policies that rely on such detailed knowledge? To address this question, we compare both resource investments and achieved confidence levels under either oMCD or the ideal control policy, for both decision scenarios (see Supplementary Methods 1 and 2 in the Supplementary Information for mathematical details regarding the derivation of the corresponding ideal policies).

We thus conducted the two following sets of Monte-Carlo simulations series. For each decision scenario, we simulate sample path trajectories of moments of value representations, under the corresponding type of value computations. Each trajectory effectively corresponds to a dummy decision trial, given some setting of the relevant decision parameters. Note that only a

**Fig. 6 | Impact of difficulty level. A** oMCD's mean resource investment (y-axis, black dots) is plotted as a function of type #1 effort efficacy (x-axis). Error-bars depict standard deviations across trials, and red diamonds show the resource investment under prospective MCD. **B** Achieved confidence, same format. **C** Achieved confidence (y-axis) is plotted against resource investments deciles (x-axis), for each difficulty level (color code: $\beta$ = type #1 effort efficacy), under oMCD's optimal policy. **D** oMCD's confidence threshold (y-axis, plain lines) is plotted against decision time (x-axis), for each difficulty level (same color code as lower-left panel). Dashed lines show expected confidence, and dots show the corresponding resource investments under prospective MCD.

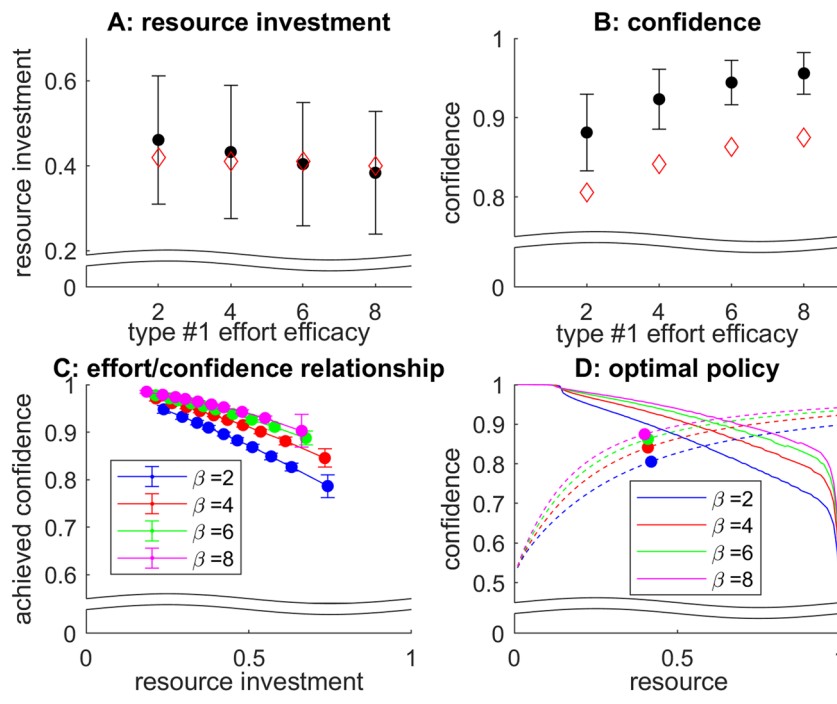

**Fig. 7 | Bayesian value denoising: comparison of oMCD and ideal control policies. A** Average resource investments under oMCD's policy (y-axis) are plotted against average resource investments under the ideal policy (x-axis), across parameter settings (dots). The color code indicates type #2 effort efficacy (blue: low $\gamma$, red: high $\gamma$). **B** Average achieved confidence, same format. **C** Average net benefit, same format.

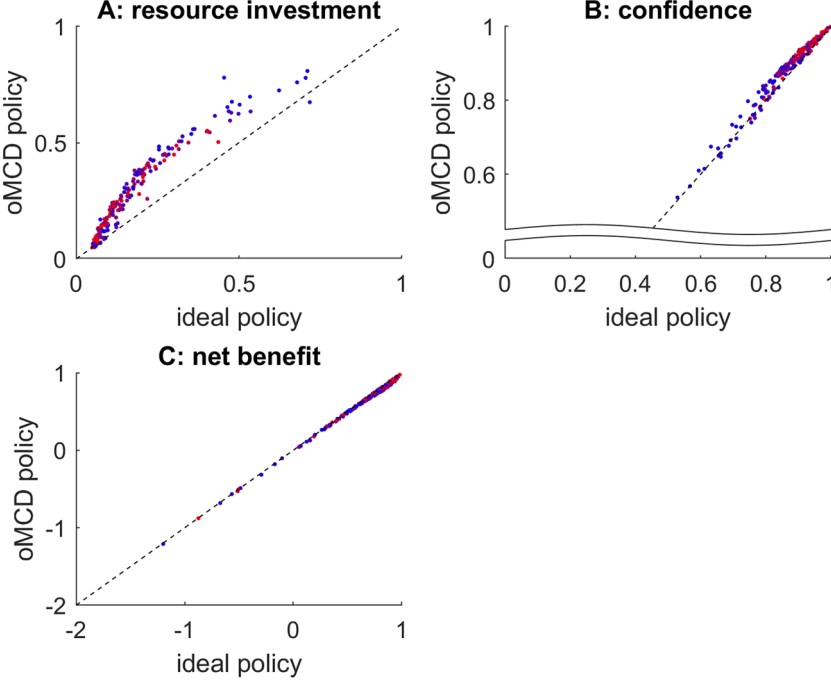

subset of these parameters is common to all decision scenarii (cost/benefit parameters, i.e.: $R$, $\alpha$ and $\nu$), whereas other parameters are typically decision-specific (*bayesian value denoising*: signal reliability $\Sigma$ and prior variance $\sigma_0$, *progressive attribute integration*: attribute moments $\eta$ and $\zeta$ as well as attribute weights $w$). For each decision parameter setting, we derive both the ideal control policy and oMCD's control policy (by approximating the effort efficacy parameters that correspond to the decision-specific parameters). We then collect the resource investments and achieved confidence that are induced by these policies, when applied on sample path trajectories of value representation moments. Now, how do ideal and oMCD policies compare across different settings of decision parameters?

Figure 7 below summarizes the comparison of ideal and oMCD policies under the *Bayesian value denoising* scenario. This comparison is made across 200 sets of randomly drawn decision parameters $\alpha$, $\nu$, $\Sigma$ and $\sigma_0$. For parameter setting, we derive the average effort investment and achieved confidence level across 500 sample path trajectories of moments of value representations.

One can see that variations in decision-relevant parameter settings induce very similar variations in average resource investments, achieved confidence and net benefits under both decision control policies. Also, although oMCD's policy yields both more effort costs (in terms of resource investments) and more benefits (in terms of achieved confidence), these

**Fig. 8 | Progressive attribute integration: comparison of oMCD and ideal control policies.** Same format as Fig. 7. The color code indicates type #1 effort efficacy (blue: low $\beta$, red: high $\beta$).

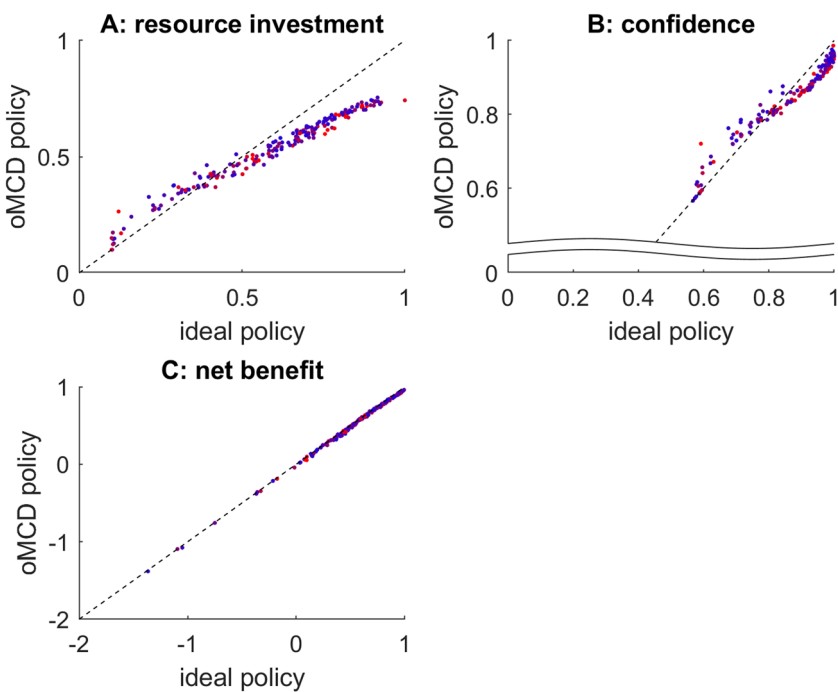

effects compensate each other and oMCD's ensuing net benefits are comparable to those of the ideal control policy. Moreover, despite oMCD's approximation of type #2 effort efficacy, it does not seem to have a systematic impact on the similarity between the two policies. These results imply that oMCD provides a tight approximation to the ideal policy for *Bayesian value denoising*.

Now Fig. 8 below summarizes the comparison of ideal and oMCD control policies under the *progressive attribute integration* scenario (200 sets of randomly drawn decision parameters $\alpha$, $\nu$, $\eta$, $\zeta$ and $w$, with $k = 10$).

As before, one can see that variations in decision-relevant parameter settings induce very similar variations in average resource investments, achieved confidence and net benefits under both control policies. Moreover, despite oMCD's approximation of type #1 effort efficacy, it does not seem to have a systematic impact on the similarity between the two policies. These results imply that oMCD provides an accurate approximation to the ideal control policy for *progressive attribute integration*.

Taken together, these results mean that the MCD architecture operates a quasi-optimal decision control that generalizes across decision processes without requiring detailed knowledge about underlying value computations.

### How critical is the definition of MCD's benefit term?

The working assumption of MCD is that decision confidence serves as the main benefit term of the resource allocation problem (cf. Equations 1–2). The advantage of this assumption is that it applies to any kind of decision process, irrespective of the underlying computations. However, as we hinted in the introduction, for the specific case of value-based decisions, there exists another natural candidate definition of the benefit term, i.e.: the value of the chosen option. One may argue that changing the definition of the benefit term effectively changes the nature of the resource allocation problem. So how critical is MCD's working assumption? Is oMCD robust to such alternative setting of the resource allocation problem?

On the computational side of things, the derivation of the ensuing optimal control policy is very similar to that of oMCD. Since the value of the chosen option is, by definition, the maximum value over the choice set, we refer to this policy as *max(value)*. It is relatively easy to show that oMCD and *max(value)* share one common important feature, i.e.: the critical quantity that triggers decisions is the absolute difference $|\Delta\mu(t)|$ in value modes.

However, in contrast to oMCD, *max(value)* is insensitive to the variance of value representations (and hence to type #1 effort efficacy). We refer the interested reader to the Supplementary Methods 3 in the Supplementary Information for mathematical details regarding the derivation of *max(value)*'s policy.

So do *max(value)* and oMCD policies respond similarly to variations in MCD parameters? To address this question, we performed the following series of Monte-Carlo simulations. First, we sample a set of MCD parameters ($\alpha$, $\beta$, $\gamma$, $\nu$ and $\kappa$) randomly. Second, we derive the optimal control threshold dynamics under both *max(value)* and oMCD policies. Third, we extract the mean response time, confidence, and net benefits over 500 random simulations of moments of value representations sample paths (according to Eq. 1). We then repeat the three steps above 200 times. The results of this analysis are summarized in Fig. 9 below.

Although oMCD tends to invest fewer resources than *max(value)* on average, it also achieves smaller confidence levels. This is essentially because the confidence mapping (cf. Equation 8) enforces an upper bound on oMCD's benefit term. Comparatively, *max(value)* thus tolerates stronger effort costs. Nevertheless, both effects compensate each other and both control policies eventually yield very similar outcomes in terms of net benefits. Unsurprisingly, each policy is (slightly) better than the other at maximizing its own benefit on average. More importantly, variations in decision parameter settings induce very similar variations in average resource investments, achieved confidence levels and net benefits. This result suggests that both frameworks are much less different than intuitively thought of, at least in terms of empirically observable decision features (choice, deliberation time, confidence). Moreover, type #1 effort efficacy, which induces variations in oMCD's policy that have no equivalent in *max(value)*, does not seem to have a systematic impact on the similarity between the two policies. In conclusion, oMCD can be thought of as providing a quasi-optimal policy for maximizing the value of the chosen option. In other terms, oMCD is robust to violations of its working assumptions.

### Does MCD reproduce established empirical results?

As we highlighted before, MCD is agnostic about the underlying decision process. However, what eventually determines the choice that is made is the inner workings of value representation updates. This is important, since some of the decision features may depend upon, e.g., whether the system

**Fig. 9 | Comparison of *max(value)* and oMCD control policies. A** Mean invested resources under oMCD's control policy (y-axis) and under *max(value)* policy (x-axis) are plotted against each other across random MCD parameter settings. The color code indicates type #1 effort efficacy (blue: low $\beta$, red: high $\beta$). **B** Mean confidence, same format. **C** Mean MCD's net benefit, same format. **D** Mean *max(value)* net benefit, same format.

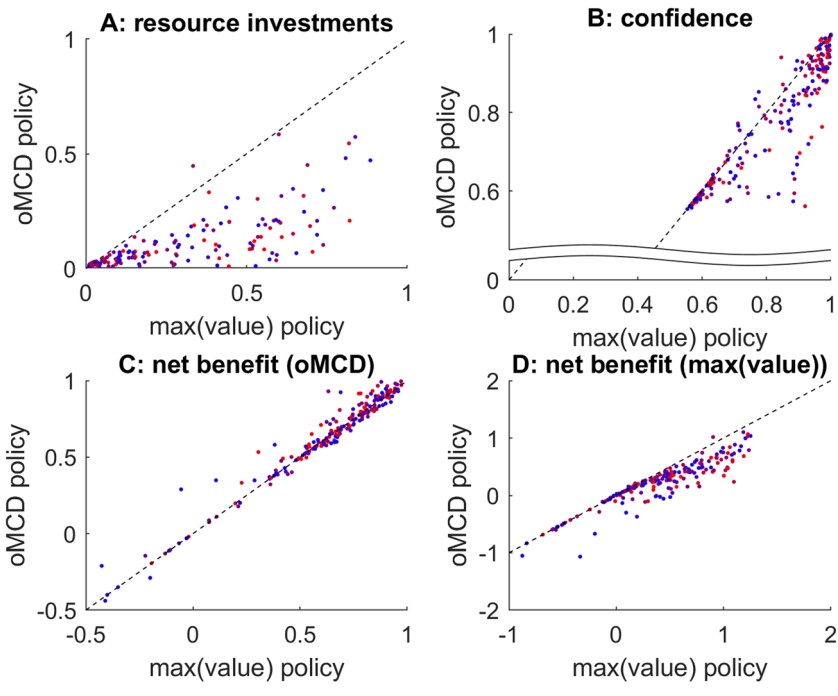

**Fig. 10 | oMCD predictions under *Bayesian value denoising*. A** The blue line and shaded area depict the mean and standard deviation of confidence trajectories (across the $10^4$ Monte-Carlo simulations), respectively. The blue dashed line shows the expected confidence under the corresponding MCD approximation, and the black dashed line shows the oMCD-optimal confidence threshold. **B** Resource investment (y-axis) is plotted against the difference in hidden option values (x-axis), for all trials (black), high-confidence trials (blue) and low-confidence trials (red), respectively. **C** The probability of choosing the first option (y-axis) is plotted against the difference in hidden option values (x-axis), for all trials (black), high-confidence trials (blue) and low-confidence trials (red), respectively. **D** Achieved choice confidence (y-axis) is plotted against the difference in hidden option values (x-axis), for all trials (black), value-consistent trials (blue) and value-inconsistent trials (red), respectively.

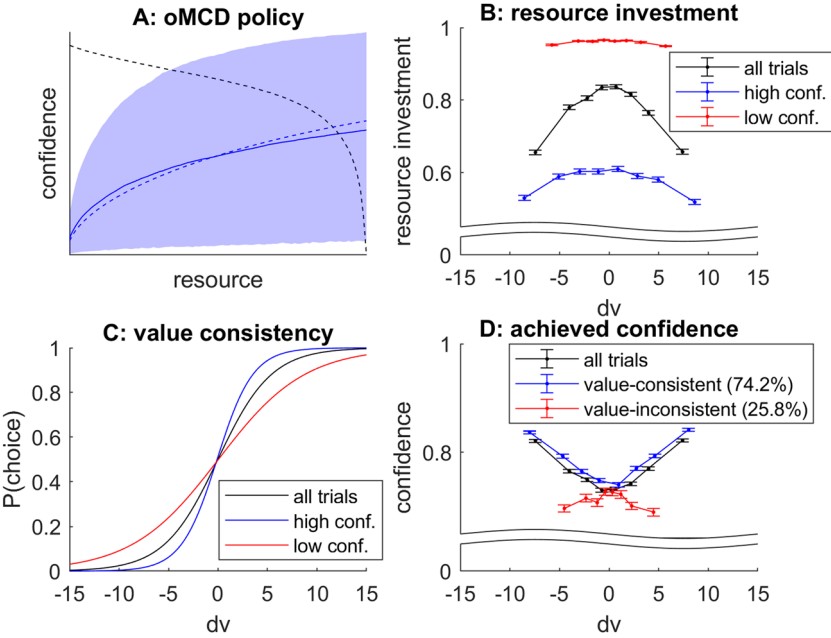

eventually arrives at a choice that is consistent with the comparison of options' values or not. Inspecting these kinds of effects thus requires performing Monte-Carlo simulations under distinct decision processes (here: *Bayesian value denoising* and *progressive attribute integration*).

Let us first consider the *Bayesian value denoising* scenario. First, we simulated $10^4$ stochastic dynamics of Bayesian value belief updates according to Eqs. 20–21, having set the decision parameters as follows: $R = 1$, $\alpha = 0.1$, $\nu = 2$, $\sigma_0 = 10$, $\mu_0 = 0$, $\Sigma = 100$, and randomly sampling trial-specific hidden value signals $V$ under the ideal observer's prior belief. Note that we chose this parameter setting because it reproduces the empirically observed rate of value-consistent/value-inconsistent decisions (see below). Second, we identified the oMCD-optimal confidence threshold dynamics, having set the effort efficacy parameters to their analytical approximation (cf. Equation 23 and related derivations). We then store the ensuing resource

investments and achieved confidence levels, as well as the choices of the decision system (as given by the comparison of value modes at decision time). Figure 10 below summarizes the results of this Monte-Carlo simulations series.

First, one can see that the MCD approximation of within-trial choice confidence dynamics is reasonably accurate (panel A), and smoothly trades errors at early and late decision times. Second, on average, resource investment decreases with the absolute difference in hidden option values (cf. black line in panel B). Third, above and beyond the effect of option value difference, resource investment decreases when choice confidence increases (cf. blue and red lines in panel B). This derives from the shape of the oMCD confidence threshold dynamics (cf. Figure 3). Fourth, the consistency of choice with value is higher for high-confidence choices than for low-confidence choices (panel C). This observation derives from performing a

logistic regression of choice against hidden value, when splitting trials according to whether they yield a high or a low level of confidence[41]. Fifth, on average, choice confidence decreases with the absolute difference in hidden option values (cf. black line in panel D). Note that the oMCD framework also predicts that confidence is higher for choices that are consistent with the comparison of hidden values than for inconsistent choices (cf. red and blue lines in panel D). This suggests that MCD possesses some level of metacognitive sensitivity[42], i.e., it reports lower confidence when making a decision that is at odds with the hidden (unknown) value. Under the assumption that decision time proxies resource investment, these are standard results in empirical studies of value-based decision making[7,13,41,43]. Interestingly, when focusing on choices that are inconsistent with the comparison of hidden values, the impact of value difference on confidence reverses, i.e., choice confidence *decreases* with the absolute difference in hidden values. This relates to known results in the context of perceptual decision making[44]. We note that these results depend upon effort cost parameters. In particular, metacognitive sensitivity tends to decrease in parameter regimes where the dynamics of oMCD confidence thresholds stop the decisions very early (e.g. low cost power and/or high unitary effort cost). This may explain the loss of metacognitive sensitivity that concurs with mental fatigue, which effectively increases one's sensitivity to cognitive effort[45].

Let us now consider the *progressive attribute integration* scenario. We essentially reproduced the same analysis as above, while simulating stochastic dynamics of value computations by attribute integration according to Eqs. 24–25, and setting the model parameters to yield a similar rate of value-consistent choices ($R = 1$, $\alpha = 3$, $\nu = 4$, $k = 20$, $\eta_k = 1$, $\varsigma_k = 1$). Figure 11 below summarizes the results of this Monte-Carlo simulations series.

In brief, one can see that we qualitatively reproduce the above relationships between effort investment, confidence and choice consistency. This is important, since this means that these relationships tend to generalize across different decision processes. However, this equivalence is only qualitative, and does not always hold. For example, reducing the unitary effort cost eventually renders the oMCD confidence threshold dynamics concave. For *progressive attribute integration*, this reverses the impact of the difference in option values onto confidence for value-inconsistent choices back again. This does not seem to happen under *Bayesian value denoising*.

For completeness, we re-analyzed the data reported in our previous investigation of (the prospective variant of) the metacognitive control of decisions[7]. In brief, participants were native French speakers, with no reported history of psychiatric or neurological illness. A total of 41 people (28 women; age: mean = 28, SD = 5, min = 20, max = 40) participated in this study (no participant was excluded). All participants rated the pleasantness of a series of food items, and performed two-alternative forced choices between pairs of (pseudo-randomly selected) items. In addition to participants' value ratings and choice, we also collected choice confidence, decision time, and subjective effort rating. We note that in this context, within-decision value computations may rely either on retrieving previously experienced food samples from episodic memory[46,47], or on integrating value-relevant attributes (e.g., tastiness and healthiness) derived from cognitive decompositions of choice options[30,48]. Both cognitive scenarios map onto *Bayesian value denoising* (which would average over memory samples) and *progressive attribute integration* processes, respectively.

We already verified the main predictions of the prospective MCD model, in terms of the relationship between pre-choice (default) value ratings and decision time/effort, as well as the ensuing decision-related variables (i.e. change-of-mind, confidence, choice-induced preference change, etc). As we already discussed, prospective and online variants of MCD make very similar predictions for these kinds of relationships. We now reproduce the above analyses (cf. Figures 10 and 11), which disclose predictions that are specific to the oMCD framework. Figure 12 below summarizes the results of these analyses.

Note that subjective effort ratings are commensurate with response times, which suggests that effort intensity shows little variations when compared to effort durations. We will comment on this in the Discussion

section below. In any case, one can see that the overall pattern of relationships between resource investments (as proxied by either decision time or reported mental effort), choice confidence and item values is qualitatively similar to that predicted from the online MCD model (cf. Figs. 10 and 11 above). Note that all the oMCD predictions discussed above are statistically significant in our empirical data:

- Effect of DV on reported effort (all trials): $t(40) = -7.6$, mean $r = -0.25 \pm 0.07$ (95% CI), $p < 10^{-4}$
- Effect of DV on reported effort (high confidence): $t(40) = -5.7$, mean $r = -0.18 \pm 0.07$ (95% CI), $p < 10^{-4}$
- Effect of DV on reported effort (low confidence): $t(40) = -5.0$, mean $r = -0.14 \pm 0.05$ (95% CI), $p < 10^{-4}$
- Effort difference (high versus low confidence): $t(40) = -7.3$, mean effort difference = $-0.19 \pm 0.05$ (95% CI), $p < 10^{-4}$
- Effect of DV on decision time (all trials): $t(40) = -7.78$, mean $r = -0.19 \pm 0.05$ (95% CI), $p < 10^{-4}$
- Effect of DV on decision time (high confidence): $t(40) = -5.9$, mean $r = -0.15 \pm 0.05$ (95% CI), $p < 10^{-4}$
- Effect of DV on decision time (low confidence): $t(40) = -3.9$, mean $r = -0.10 \pm 0.05$ (95% CI), $p = 0.0002$
- Response time difference (high versus low confidence): $t(40) = -7.0$, mean RT difference = $-0.62 \pm 0.17$ (95% CI), $p < 10^{-4}$
- Effect of DV on choice (all trials): $t(40) = 25.2$, mean effect size = $1.56 \pm 0.12$ (logistic regression, 95% CI), $p < 10^{-4}$
- Effect of DV on choice (high confidence): $t(40) = 32.6$, mean effect size = $2.02 \pm 0.12$ (logistic regression, 95% CI), $p < 10^{-4}$
- Effect of DV on choice (low confidence): $t(40) = 10.4$, mean effect size = $0.84 \pm 0.16$ (logistic regression, 95% CI), $p < 10^{-4}$
- Effect of DV on choice (high versus low confidence): $t(40) = 13.8$, mean effect size difference = $1.17 \pm 0.16$ (logistic regression, 95% CI), $p < 10^{-4}$
- Effect of DV on confidence (all trials): $t(40) = 8.5$, mean $r = 0.27 \pm 0.06$ (95% CI), $p < 10^{-4}$
- Effect of DV on confidence (value-consistent): $t(40) = 10.6$, mean $r = 0.27 \pm 0.05$ (95% CI), $p < 10^{-4}$
- Effect of DV on confidence (value-inconsistent): $t(40) = -4.22$, mean $r = -0.18 \pm 0.09$ (95% CI), $p < 10^{-4}$
- Confidence difference (value-consistent versus value-inconsistent): $t(40) = 10.8$, mean confidence difference = $0.10 \pm 0.02$ (95% CI), $p < 10^{-4}$

where DV stands for difference in option values, all statistical significance tests are one-sided and derive from standard random effect analyses (sample size: $n = 41$). We note that these analyses were not part of a preregistration protocol.

## Discussion

In this work, we have presented the online/reactive metacognitive control of decisions or oMCD framework.

### Limitations

To begin with, recall that we have framed oMCD as a solution to a resource allocation problem. More precisely, we think of decision deliberation as involving the investment of costly cognitive resources, which are necessary to process decision-relevant information. The outcome of such resource allocation is to override default behavioral responses, which would otherwise be triggered by automatic (e.g., reflexive, habitual or intuitive) brain processes. Under this view, the brain faces the problem of adjusting *the amount* of resources to invest, which we equate with the issue of effort regulation. This perspective is not novel: the notion of mental effort was central to the early definition of automatic versus controlled processing, with the former described as quick and effortless, and the latter as slower and effortful[49]. Since controlled processes are slow, it is reasonable to assume that the brain may regulate effort simply by adjusting its duration. This is the premise of our computational framework, which relies on the theory of

**Fig. 11 | oMCD predictions under progressive attribute integration.** Same format as Fig. 10.

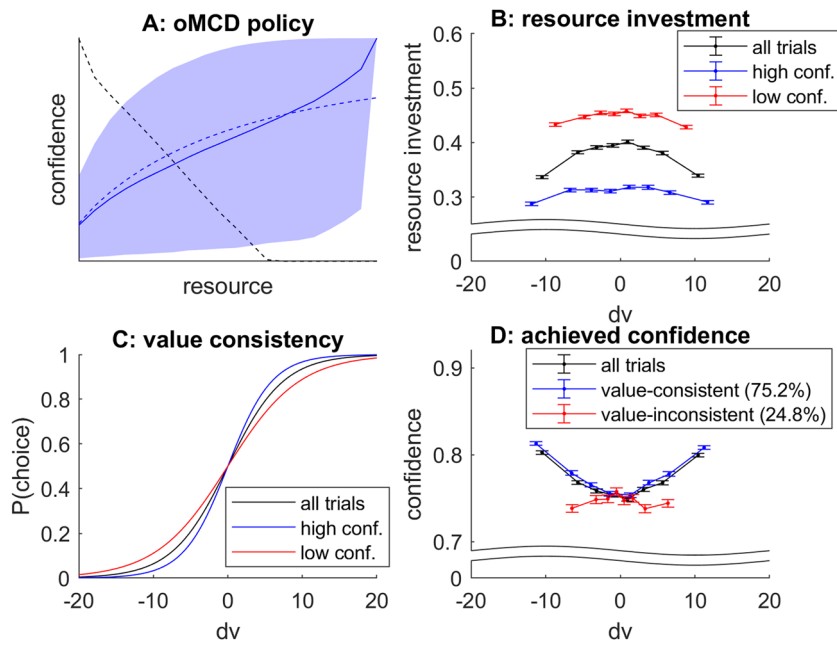

**Fig. 12 | Re-analysis of behavioral data in a simple value-based decision making experiment[7].**
**A** Reported mental effort (y-axis) is plotted against the difference in reported option values (x-axis), for all trials (black), high-confidence trials (blue) and low-confidence trials (red), respectively. **B** Response time, same format. **C, D** Same format as Fig. 10.

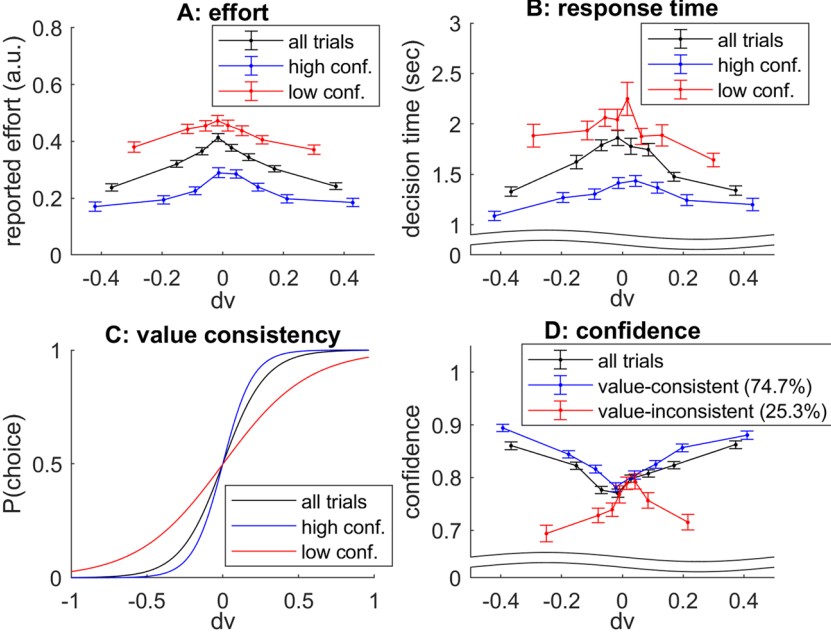

optimal stopping[21]. However, effort actually unfolds along two dimensions: duration and intensity. This means that, in principle, both decision speed and confidence may be increased at the cost of increasing effort intensity. Accordingly, investing cognitive control is known to speed up responses in the context of, e.g., behavioral conflict tasks[50,51]. This raises the question: what determines the brain's policy for trading effort intensity against effort duration? A possibility is that this depends upon the nature of the cognitive resource that is required for processing decision-relevant information. The issues of *how* to control resource investment and *which* resource to invest are thus intertwined[2]. For example, one may think of resources as being composed of cognitive modules, such as working memory or attention, whose neurobiological underpinnings may induce distinct costs and/or limitations on effort intensity and duration[52–54]. More generally, the effort intensity/duration tradeoff may be eventually determined by the neuro-biological constraints that are imposed on the neural architecture that

operates the processing of decision-relevant information[4,55]. For example, value-based decision making may require the active maintenance of multiple value representations that tend to interfere with each other, e.g., because they involve the same neural population within the orbitofrontal cortex[32]. In this case, cognitive control may alter the OFC neural code with the aim of temporarily dampening these interferences. In principle, the associated neural mechanism may operate based on simple confidence monitoring (which would proxy value conflict signals), without knowledge of the intricate architecture of value coding in the OFC. We will test these ideas using artificial neural network models of MCD in forthcoming publications.

**On the generality of oMCD control policy**
One of the main assumptions behind MCD is that mental effort investment is regulated by a unique controller that operates under agnostic assumptions

about the inner workings of the underlying decision system. This constraint somehow culminates in the simplicity of oMCD's control architecture, which reduces to a monitoring of decision confidence. In this context, we have shown that the optimal stopping policies of distinct decision processes (*Bayesian value denoising* or *progressive attribute integration*) can be approximated using a simple calibration of effort efficacy parameters. We have also highlighted the ensuing properties of oMCD: when coupled with these different underlying decision systems, oMCD reproduces most established empirical results in the field of value-based decision-making. In addition, we have shown that oMCD is robust to alternative settings of the resource allocation problem. In particular, decision confidence seems to be a reasonable proxy for the value of the chosen option, which is the standard candidate titration for the benefit of value-value based decisions[14,24]. Taken together, these results suggest that the architecture of oMCD control, which relies on the internal monitoring of decision confidence, may generalize to most kinds of decision processes. Preliminary investigations show that this holds for yet another important kind of value-based decisions, whereby value computation is the output of a forward planning process on a decision tree[56,57]. Arguably, this also holds for perceptual or evidence-based decisions. In this context, decision confidence can be defined - somewhat more straightforwardly - as the subjective probability of being correct[35]. As long as effort efficacy parameters can be simply identified, the MCD architecture will provide an accurate approximation to the optimal resource allocation policy. This is trivial when perceptual detection or discrimination processes can be described as some form of *Bayesian denoising* of some perceptual variable of interest[23,40]. This would also hold for perceptual categorization processes, which may rather resemble *attribute integration* scenarios[19]. In fact, oMCD's potential generalizability derives from its agnostic stance regarding the nature of information processing that takes place in the underlying decision system. This is also why oMCD can in principle be extended to describe the metacognitive control of other kinds of cognitive processes (e.g., reasoning or memory encoding/retrieval). In this context, an interesting avenue of investigation would be to consider the impact of metacognitive adaptation on the generalization of control policies across cognitive domains. Note that, because we assume MCD's control architecture to be invariant across contexts, it requires a systematic calibration (in terms of, e.g., effort costs and/or efficacies) to guaranty the quasi-optimality of resource allocation. As we highlighted before, we expect such calibration to converge very quickly (e.g., over a few training trials). This is because effort efficacies can be learned from within-trial confidence dynamics. Nevertheless, whether this specific kind of metacognitive adaptation is sufficient to recycle and adjust MCD's control architecture to novel cognitive domains, as well as how it shapes cross-domain metacognitive learning effects, is virtually unknown and would require specific empirical tests.

## On the difference between prospective and online/reactive variants of MCD

Retrospectively, prospective and online/reactive variants of MCD solve the same computational problem, i.e. maximizing the expected net benefit of resource allocation. We have shown that their respective control policies share many common features. In particular, they tend to respond similarly to changes in effort costs and/or efficacies. However, they differ in at least two important aspects. First, although its algorithmic derivation is more sophisticated, oMCD's control policy is computationally simpler than its prospective variant. This is because it does not require an explicit comparison of all admissible resource investments prior to decision deliberation. Rather, it relies on dynamical changes in decision confidence signals to trigger a binary (yes/no) stopping decision. In other terms, the comparison between admissible resource investments is performed implicitly, while the control system monitors the progress of the underlying decision system. This renders the neurocomputational architecture of oMCD very similar to basic Drift Diffusion Decision Models or DDMs, whose candidate neural underpinnings have been partially identified[58–60]. Second, only oMCD predicts non trivial second-order statistics on key decision features beyond those induced by changes in effort costs and efficacies. For example, both prospective and online/reactive MCD typically predict a negative correlation between reported confidence levels and response times *across difficulty levels* (as induced by different type #1 effort efficacies), but only oMCD predicts such a relationship *within each difficulty level* (across trials). The range and diversity of non trivial second-order statistics that oMCD predicts is exemplified in Figs. 10, 11. We note that some of these predicted statistical relationships are within the grasp of those existing variants of DDMs that explicitly account for decision confidence. This holds, e.g., for the two-way interaction between confidence and item values onto response time and choice[41]. Others may be more specific to oMCD (and related ideal control policies), e.g., the inversion of the value/confidence relationship for value-consistent and value-inconsistent choices. In any case, these non trivial second-order statistics are the hallmark of online/reactive control policies. In this context, what oMCD offers is a way to predict how these relationships should change, would effort costs and/or efficacies be experimentally manipulated.

## On extending MCD with goal hierarchies

Whether MCD is operated online or not, it relies upon some prospective computation, which anticipates the costs and benefits of investing additional resources in the decision. In turn, the optimal cost-benefit tradeoff relies upon decision-specific features, such as decision importance and difficulty. The former is signalled by the weight parameter $R$ that scales confidence in the benefit term (cf. Equation 1). In our previous empirical work on MCD, participants were asked to decide between pairs of food items. In this context, we manipulated decision importance by instructing participants that they would have to eat the item they eventually chose (so-called "consequential decisions") or not. As predicted by the MCD framework, increasing decision importance systematically increases decision time, above and beyond the effect of option values[7]. In other terms, increasing decision importance may overcompensate the cost of mental effort by increasing the demand for confidence. More generally, we think of $R$ as the expected reward attached to the attainment of the superordinate goal, within which the decision is framed. Importantly, although $R$ is analogous to a reward, it is distinct from the values that are attached to the choice options. This does not mean that the values that decision systems attach to choice options are independent from the goal: recent research has demonstrated that option values are strongly influenced by how useful choice options are for achieving one's goal[12,61]. However, at least in principle, alternative choice options that would be instrumental for attaining an important goal may still have low value. For example, while starving, one may only have access to low quality/palatability food items. A possibility is to conceive of goals as being organized hierarchically, whereby superordinate goals are broken down into candidate subordinate goals[7,62,63]. According to MCD, the selection of subordinate goals would be under higher scrutiny when superordinate stakes increase (everything else being equal). Having said this, the urgency of attaining superordinate goals may also incur additional temporal costs for subordinate goal selection, which may overcompensate the increased demand for confidence (as would be the case for, e.g., starvation). We intend to investigate these kinds of issues in forthcoming publications.

## Data availability
All empirical data (as well as analysis code) is available here: https://owncloud.icm-institute.org/index.php/s/wAsSPNndwZVlBlR.

## Code availability
The matlab code that was used to generate all Figures in this manuscript is available here: https://owncloud.icm-institute.org/index.php/s/nXnbv2b3gtNz0Jj. This code is also available as part of the VBA academic freeware (https://mbb-team.github.io/VBA-toolbox/), which is versioned and regularly updated.

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

## Acknowledgements

This work was realised with the following funding: ICM-IHU Big Brain Theory Project (BBT2016-decimotiv) and Agence Nationale de la Recherche (ANR-20-CE37-0006). The funders had no role in study design, data collection and analysis, decision to publish or preparation of the manuscript.

## Author contributions

Douglas Lee collected the empirical data which we re-analyze in this work. Juliette Benon, Douglas Lee and Jean Daunizeau derived the mathematical model and analyzed the empirical data. Juliette Benon, Douglas Lee, William Hopper, Morgan Verdeil, Mathias Pessiglione, Fabien Vinckier, Sebastien Bouret, Marion Rouault, Raphael Lebouc, Giovanni Pezzulo, Christiane Schreiweis, Eric Burguière and Jean Daunizeau contributed to elaborating the oMCD theoretical framework and wrote the paper.

## Competing interests

The authors declare no competing interests.
