## [Peer Review File · Communications Psychology]

14th Jun 23

Dear Jean,

Thank you for your patience during the peer-review process. Your manuscript titled "The online metacognitive control of decisions" has now been seen by 3 reviewers, and I include their comments at the end of this message. They find your work of interest, but raised some important points. We are interested in the possibility of publishing your study in Communications Psychology, but would like to consider your responses to these concerns and assess a revised manuscript before we make a final decision on publication.

We therefore invite you to revise and resubmit your manuscript, along with a point-by-point response to the reviewers. Please highlight all changes in the manuscript text file.

Reviewers #1 and #2 are subject-matter experts who provided feedback on the conceptual and methodological aspects of the work. Reviewer #3 is likewise a subject-matter expert, but in the present case, was invited to undertake a verification of the functionality of the model code and therefore not required to comment on the manuscript itself.

We ask that you address all presentational issues raised by Reviewers #1, #2, provide control analysis as necessary to alleviate the concerns expressed by Reviewer #2. Finally, please ensure that all issues related to the code identified by Reviewer #3 are resolved, to maximize reproducibility and utility of the deposited model.

To allow us to process your manuscript more swiftly following revisions, we strongly recommend ensuring that it fully complies with our checklist: Communications Psychology formatting checklist at the point of resubmission.

For future reference, please be advised that when data or code are made available for peer-review purposes, the nature portfolio's policy requires that referees can access these materials without disclosing any personal information.

Please use the following link to submit your revised manuscript, point-by-point response to the referees' comments (which should be in a separate document to any cover letter) and the completed checklist:

[link redacted]

We hope to receive your revised paper within 4 weeks; please let us know if you aren't able to submit it within this time so that we can discuss how best to proceed. If we don't hear from you, and the revision process takes significantly longer, we may close your file. In this event, we will still be happy to reconsider your paper at a later date, provided it still presents a significant contribution to the literature at that stage.

Please do not hesitate to contact me if you have any questions or would like to discuss these revisions further. We look forward to seeing the revised manuscript and thank you for the opportunity to review your work.

Best wishes,

Marike

Marike Schiffer, PhD
Chief Editor
Communications Psychology

EDITORIAL POLICIES AND FORMATTING

Editorial Policy: Policy requirements (Download the link to your computer as a PDF.)

Furthermore, please align your manuscript with our format requirements, which are summarized on the following checklist:

Communications Psychology formatting checklist

and also in our style and formatting guide Communications Psychology formatting guide .

* **CODE AVAILABILITY:** All Communications Psychology manuscripts must include a section titled "Code Availability" at the end of the methods section. In the event of publication, we require that the custom analysis code supporting your conclusions is made available in a publicly accessible repository; at publication, we ask you to choose a repository that provides a DOI for the code; the link to the repository and the DOI will need to be included in the Code Availability statement. Publication as Supplementary Information will not suffice. We ask you to prepare code at this stage, to avoid delays later on in the process.

* **DATA AVAILABILITY:**

All Communications Psychology manuscripts must include a section titled "Data Availability" at the end of the Methods section or main text (if no Methods). More information on this policy, is available at <http://www.nature.com/authors/policies/data/data-availability-statements-data-citations.pdf>.

At a minimum the Data availability statement must explain how the data can be obtained and whether there are any restrictions on data sharing. Communications Psychology strongly endorses open sharing of data. If you do make your data openly available, please include in the statement:

We recommend submitting the data to discipline-specific, community-recognized repositories, where possible and a list of recommended repositories is provided at <http://www.nature.com/sdata/policies/repositories>.

If a community resource is unavailable, data can be submitted to generalist repositories such as figshare or Dryad Digital Repository. Please provide a unique identifier for the data (for example a DOI or a permanent URL) in the data availability statement, if possible. If the repository does not provide identifiers, we encourage authors to supply the search terms that will return the data. For data that have been obtained from publicly available sources, please provide a URL and the specific data product name in the data availability statement. Data with a DOI should be further cited in the methods reference section.

REVIEWERS' EXPERTISE:

Reviewer #1: decision-making, computational modelling

Reviewer #2: decision-making, computational modelling

Reviewer #2: Code review (decision-making, computational modelling)

REVIEWERS' COMMENTS:

Reviewer #1 (Remarks to the Author):

In the present paper, the authors expand a prior model on effort allocation in decision-making. Like the original model, this new oMCD model trades effort and confidence, where effort is a cost term and confidence is a benefit term in a cost-benefit decision. Unlike the original model which prospectively determines how much effort to invest (i.e., how long to sample options before committing to a response), oMCD updates this decision online, i.e., in a reactive manner. Specifically, rather than prospectively estimating the relevant components from an early data sample, the expected benefit of effort is iteratively updated contingent on the momentary value estimates and crucially their difference.

The authors show that this model is optimal in two different decision scenarios: for optimal stopping, as well as multi-attribute choice. They also show that it reproduces typical empirical findings in both scenarios.

Comparing both models, the oMCD model outperforms its prospective predecessor across a range of indices and approaches the performance of an omniscient oracle model in which all relevant parameters are known. That is, on average the model reaches greater confidence at lower effort investments, maximizing the discounted benefit.

Intriguingly, oMCD makes novel predictions and can generate a wider range of RT-Confidence relationships because of its inherent parameter dependence and dynamics.

I believe this is an interesting paper and that the model has great potential in explaining behavior. I thought that for a very mathematical paper, the writing was very accessible and the authors did a good job contextualizing and unpacking their work. My only concern is for the discussion.

The discussion is very focussed on future directions. That is it very quickly jumped into effort costs, effort intensity versus time, and generalizeability, all of which seem like they're not the most immediate things to discuss about the oMCD specifically. Meanwhile the authors have room to unpack the differences in the predictions the MCD model (and perhaps other normative models) and the oMCD model make, how the new oMCD predictions could allow us to understand complex empirical effects they allude to in the introduction, such as context dependent relationships between RT and confidence. I believe the discussion could use a bit more unpacking on that end before diving into the very general questions that are currently featured and addressed by a much wider set of literature than decision-making per se.

Minor:

In figures, e.g. figure 6 where relationships between variables such as confidence and rt are shown, maybe put those on the axis labels in addition to or instead of the formulas. That's hard to read otherwise.

Reviewer #2 (Remarks to the Author):

Summary

The manuscript presents an expanded version of a model for how people decide how much effort to invest in decisions. The focus is on value-based decisions, but as noted in the manuscript, the framework can be directly transferred to other kinds of decision. The extension is to consider a model that can reevaluate mid-trial how much time to spend deliberating (online MCD), as opposed to simply deciding before a trial begins, how much time to spend (prospective MCD). This seems like an important advance over the existing model. A very nice feature of the new model is that it is approximately valid for a range of specific decision-making mechanisms, as elaborated in the manuscript. I found these unifying derivations very innovative, and I find it an exciting line of work because it could support the parsimonious explanation of psychological phenomena in a range of tasks with a single model.

Overall, I think the manuscript offers a valuable contribution to the literature on balancing benefits and resource costs in decision-making, but I had some concerns which I think first need to be addressed. I list bigger points first and then smaller points.

Cognitive resources

I am not convinced that the framing of the paper accurately describes the manuscript's contribution. The stated aim of the manuscript is to explore control and specifically resource allocation. However, in the new online version of MCD, "resource" takes on a very limited definition of being equal to the time spent deliberating. The question for the observer is no longer how much resources to allocate, but how much time to spend deliberating. In the discussion the authors acknowledge that it is theoretically possible that observers spend more or less resources per unit time (through the parameter κ), but this possibility is not explored in the manuscript. Therefore, I am not sure that it is fair to say that the oMCD model is a model of the deployment of cognitive resources, when the only "resource" that observers have control over is decision time. I wonder whether a more accurate framing would be to present the model as a model of deliberation time.

Benefit term

The models all concern how observers trade cost for benefit. The models assume that the "benefit" that observers are aiming to maximise is confidence. The motivation behind this choice is not clear to me because, in the context of value-based decision making, there is a very clear and face-valid way to operationalise "benefit". Specifically, "benefit" can simply be operationalised as obtained reward or obtained value. Probably because it is so straightforward, maximising reward/value has been the standard goal in previous work (e.g. Tajima et al., 2019a). My question is, what is the motivation for departing from this face-valid and standard measure of "benefit"?

Nature of the approximation

The manuscript nicely shows how different decision-making scenarios can be approximated under a more general model. I found these fascinating and very innovative results. In the manuscript it is argued that observers use this approximation because the relevant "controller" does not have knowledge of decision-making processes. The assumption that the controlling process has no knowledge of decision-processes is tenable, but it strikes me as a very strong assumption that deserves at least some justification or motivation. In particular, the idea that the controller has limited knowledge of the mechanisms of decision-making seems in tension with the common assumption that observers can use normative decision strategies by learning the statistical properties of decision-making tasks: If observers can learn the statistics of external stimuli, it would seem sensible to assume that they could also learn the statistics of their own decision-making mechanisms.

Testing / justifying approximations

Substantial approximations were made in the derivations of oMCD for the "ideal observer case" and "attribute-integration case". In particular I am thinking about the approximations on Ln 427 and Ln 489. However, as I understand the manuscript, the effect of these approximations was not directly studied or tested. For example, we do not know how much worse oMCD performs than the optimal solution for the two cases.

The effect of some of these approximations could be substantial. Consider the approximation on Ln 426. I assume that the range of admissible times is very big, i.e. the final admissible time, T , is very large (to ensure that this artificial deadline has little practical effect in the model). In this case, won't the average in Ln 427 be dominated by very large values of t , which are hardly ever realised? I.e. the average is over all admissible times, but most admissible times are never actually reached, with decisions usually happening much earlier. In this case, isn't the average quite a poor approximation? More generally, I wondered whether the authors had explored the effect of this approximation.

Making assumptions clear

Sometimes assumptions were not clearly explained for the reader. Assumptions are of course crucial to any derivation, but it should be clear to the reader, what assumptions were made, and the ways in which they limit the scope of the derivations. For example, the text states (Ln 245) "Without loss of generality, we assume that there is a linear relationship between deliberation time and resource investment, i.e.: $z = kt$ ". But this is a strong loss of generality: The relationship could have been piecewise-linear or could have been any non-linear form. All these possibilities are being ruled out. To be clear, I am not asking the authors to reconsider which assumptions they made, just to make small changes to how they are presented.

Other examples:

- Ln 149 – 151 is another place where the text states "without loss of generality", but an assumption is made that reduces generality.
- Ln 176: "this implies that the precision of a given probabilistic value representation necessarily increases in proportion to the amount of allocated resources". What is necessary about this? Isn't in the case that various relationships would be possible (quadratic, generic monotonic, e.c.t.). This is a sensible assumption, but I don't see how it is "necessary".

In contrast, I felt the assumptions behind equation 4 (Ln 190 – 200) were introduced in a very clear, well-motivated manner.

Detailed points

- It could improve readability if the subplots were labelled A, B, C... and these referred to in the text, when a specific subplot in a figure is relevant.
- Ln 87 "kind of problems"  "kind of problem"?
- Ln 96 "Interestingly, authors"  "Interestingly, the authors"?
- Ln 98 Could it not be the case that the optimal threshold decreases very slowly, and hence a detectable decrease in confidence with time in empirical data is not necessarily expected?
- Ln 100: There is also uncertainty regarding decision difficulty in Tajima et al., (2019a), isn't there? As I remember it, the strength of the value signal can vary, hence variability in decision difficulty. In any case, my understanding is that uncertainty regarding decision difficulty often generates optimal decision boundaries that decrease with time (E.g. Fig. 3 in Malhotra et al., 2018), which seems inconsistent with the argument being put forward here.
- Ln 106 "confidence reports as already been"  "confidence reports has already been". Please check the text throughout for typographical errors.
- Ln 128. Is the use of the word "disclaimer" necessary? It sounds very legal.
- Ln 147 is somewhat unambiguous. It sounds as if V_i denotes "internal representations of values",

but from what I understand V_i denotes the real external values, and the internal representation of those real external values are what is probabilistic

- Ln 315. From this line it is clear that “a” and “ π ” should be coded in the same way, but I believe the coding is inconsistent. Ln 269 says $a=0$ means the system stops. From Ln 281 I understand that $\pi=0$ means the system continues. Please correct me if I am wrong.
- Ln 309 / Equation 14. (A) I understood from above that $a=1$ means wait, but in the equation here (as I understand it) the $a=1$ case gives the reward associated with stopping. (B) From Equation 9 we have that $Q(\dots)=0$ if $a=1$. Hence, shouldn't at least one of the cases in Equation 14 be 0 (specifically the case associated with waiting).
- Ln 382. I find “ideal observer case” a very confusing name for this case because, whatever information the observer receives, there will be some optimal way to process that information. For example, if the observer only receives information on how two options differ on specific attributes, there will still be an optimal way to use this attribute information that could then be described as an “ideal observer”. This case currently labelled “ideal observer” includes very specific assumptions (eg. Ln 389) which don't have any necessary connection to an ideal observer, but are rather substantive assumptions about decision-making mechanisms, and what information is received by them. Perhaps a better name would be something like “Gaussian noise case”.
- Ln 586 – 587. What is the reason for excluding these no deliberation / no resource investment cases? They are still legitimate cases aren't they, and the model's strategy of no deliberation possibly normative?
- Ln 700. “decreases”  “increases”?
- Ln 844: An approximately optimal solution, or an optimal solution?

Reviewer #3 (Remarks to the Author):

This is a review of the functionality of the code as per the invitation instructions: “to verify that the code included in this study is functional, that it is described in sufficient detail, and that it reproduces the results reported in the study”.

The supplementary materials for this manuscript contains the details to download code for this paper, and the toolbox used in that code, and details of running some demo scripts. The equations underlying the code are described appropriately in the manuscript, along with the steps taken in each simulation.

However, I cannot find the data/code availability statement in the main manuscript (as the Editorial Policy Checklist states is done), nor any reference to the software/language/toolboxes used for the simulations. This information is given in the reporting summary documents, but should be stated in the main manuscript also.

I could not reproduce all of the figure panels, as two scripts appeared to be missing from either the VBA toolbox or the oMCD code (VBA_plotxy.m and VBA_logisticReg.m, see error messages below). I could not find any reference to these scripts online at all. The VBA

Aside from those errors, I could reproduce all other figure panels, apart from Figure 1 which does not appear to have any code given in the readme file.

The download and installation works fine (though the VBA toolbox download they offer requires name & email in order to download from the authors website, which should not be required for blinded peer review, so perhaps this journal's code requirements should not allow this in the future).

I have detailed the steps I took, and all warnings/errors I encountered below. I have run each demo to produce the figures in the paper, using Matlab R2021b on Windows 10.

Some scripts took far longer than the stated 1 minute to run, due to the warnings produced by using a deprecated function in the VBA toolbox (see warnings below). Manually turning off these warnings greatly sped up the scripts to well below 1 minute (e.g. demo_MCDs.m took way over 20 minutes to run, which became 32 seconds after commenting those warnings out). It seems like the functions 'sig.m' and 'vec.m' are wrapper functions to call 'VBA_sigmoid.m' and 'VBA_vec.m' and print the warning, so they should probably be replaced with those newer scripts throughout the oMCD code.

Warning: *** The function `sig` is now deprecated. Please see `VBA_sigmoid` for an alternative.

> In sig (line 4)

In getPc>myEabs2 (line 32)

In getPc (line 19)

In demo_MCDs (line 71)

Warning: *** The function `vec` is now deprecated and has been renamed `VBA_vec`.

> In vec (line 4)

In get_wMDP0 (line 23)

In demo_MCDs (line 53)

I ran each demo in the readme:

1. No code is presented to replicate Figure 1
2. demo_Econf.m runs fine (these figures are not in the paper)
3. demo_onlineMCD0.m reproduces figure 2, but then gave an error during plotting Figure 3d (panels a-c are reproduced though), as it appears the function 'VBA_plotxy' is not included in either the toolbox or the code (see error message below). Figure 4 is reproduced if the cell after this error is run.
Unrecognized function or variable 'VBA_plotxy'.
Error in demo_onlineMCD0 (line 168)
out = VBA_plotxy(t_omcd,P_omcd,10,ha,0);
4. demo_MCDs.m ran fine and reproduced figure 5 (plus an additional panel which does not appear in the manuscript)
5. demo_oMCD_difficulty.m reproduced Figure6a-b but gave the 'VBA_plotxy' error (see above point #3 above, occurred on line 114 in the current script), so could not make panel C. Panel d was made if the code after that error was run (although the legend did not contain all the different effort level labels, due to that error halting the code). Figure 7 can be reproduced by changing all the parameters at the top to 2 (apart from panel C due to this missing script).
6. demo_oMCD_thresholds.m reproduces Figure 8 (as a side note, I would recommend moving the legends in Figure 8a+b to not cover the magenta line).
7. demo_oMCD_Vupdates.m again has the VBA_plotxy error (line 90), as well as a missing function VBA_logisticReg.m (line 106), so I am only able to reproduce panels a of Figures 9 and 10

Title : The online metacognitive control of decisions

Journal: Communications Psychology

Manuscript id: COMMSPSYCHOL-23-0088-T

To begin with, we would like to thank the reviewers for giving us the opportunity to improve this work, and we apologize for the time it took us to revise it.

We have made substantial modifications to the previous version of our manuscript, which we briefly summarize below:

- We have derived the ideal stopping policies for the two decision scenarios that exemplify, in our work, the diversity of value computations. We have compared oMCD to these policies, eventually demonstrating that it provides a tight approximation, despite being agnostic on the underlying value computations.
- We have derived the optimal policy for an alternative framing of the resource allocation problem, whereby the benefit term is defined as the value of the chosen option. Here again, we show that oMCD provides a tight approximation to this policy.
- Accordingly, we have reported the above two sets of analyses into the revised Results section (and have inserted three new Appendix sections to summarize the associated mathematical derivations). For the sake of conciseness, we have also removed some of the earlier reported results, which were not essential to this work.
- We have modified the Introduction section to better motivate oMCD's working assumption regarding the use of confidence for titrating the benefit of resource allocation.
- We have improved the Model section, in the aim of clarifying our assumptions and derivations.
- We have entirely rewritten the Discussion section, eventually focusing on the most important issues raised during this review round.
- We have resolved all reported code issues

Note that we have reversed the order of the first two authors of the manuscript, given the effort investment of Juliette Benon (previously author #2) into the manuscript revisions.

We hope that you will find that our revised manuscript addresses your comments and concerns.

Reviewer #1 (Remarks to the Author):

In the present paper, the authors expand a prior model on effort allocation in decision-making. Like the original model, this new oMCD model trades effort and confidence, where effort is a cost term and confidence is a benefit term in a cost-benefit decision. Unlike the original model which prospectively determines how much effort to invest (i.e., how long to sample options before committing to a

response), oMCD updates this decision online, i.e., in a reactive manner. Specifically, rather than prospectively estimating the relevant components from an early data sample, the expected benefit of effort is iteratively updated contingent on the momentary value estimates and crucially their difference.

The authors show that this model is optimal in two different decision scenarios: for optimal stopping, as well as multi-attribute choice. They also show that it reproduces typical empirical findings in both scenarios.

Comparing both models, the oMCD model outperforms its prospective predecessor across a range of indices and approaches the performance of an omniscient oracle model in which all relevant parameters are known. That is, on average the model reaches greater confidence at lower effort investments, maximizing the discounted benefit.

Intriguingly, oMCD makes novel predictions and can generate a wider range of RT-Confidence relationships because of its inherent parameter dependence and dynamics.

I believe this is an interesting paper and that the model has great potential in explaining behavior. I thought that for a very mathematical paper, the writing was very accessible and the authors did a good job contextualizing and unpacking their work. My only concern is for the discussion.

We thank you for your positive evaluation of our work!

The discussion is very focussed on future directions. That is it very quickly jumped into effort costs, effort intensity versus time, and generalizeability, all of which seem like they're not the most immediate things to discuss about the oMCD specifically. Meanwhile the authors have room to unpack the differences in the predictions the MCD model (and perhaps other normative models) and the oMCD model make, how the new oMCD predictions could allow us to understand complex empirical effects they allude to in the introduction, such as context dependent relationships between RT and confidence. I believe the discussion could use a bit more unpacking on that end before diving into the very general questions that are currently featured and addressed by a much wider set of literature than decision-making per se.

We agree that oMCD's properties deserves more unpacking... and that discussing effort cost theories is somehow far-fetched in this context. First, we have modified the Results section to improve on this aspect. Second, we have included the following paragraph in the Discussion section (and have removed the Discussion about effort costs, etc):

Retrospectively, prospective and online/reactive variants of MCD solve the same computational problem, i.e. maximizing the expected net benefit of resource allocation. We have shown that their respective control policies share many common features. In particular, they tend to respond similarly to changes in effort costs and/or efficacies. However, they differ in at least two important aspects. First, although its algorithmic derivation is more sophisticated, oMCD's control policy is computationally simpler than its prospective variant. This is because it does not require an explicit comparison of all admissible resource investments prior to decision deliberation. Rather, it relies on dynamical changes in decision confidence signals to trigger a binary (yes/no) stopping decision. In other terms, the comparison between admissible resource investments is performed implicitly, while the control system monitors the progress of the underlying decision system. This renders the neurocomputational architecture of oMCD very similar to basic Drift Diffusion Decision Models or DDMs, whose candidate neural underpinnings have been partially identified (Gold & Shadlen, 2007; Lam et al., 2022; Turner et al., 2015). Second, only oMCD predicts non trivial second-order statistics on key decision features beyond those induced

by changes in effort costs and efficacies. For example, both prospective and online/reactive MCD typically predict a negative correlation between reported confidence levels and response times *across difficulty levels* (as induced by different type #1 effort efficacies), but only oMCD predicts such a relationship *within each difficulty level* (across trials). The range and diversity of non trivial second-order statistics that oMCD predicts is exemplified in Figures 10-11. We note that some of these predicted statistical relationships are within the grasp of those existing variants of DDMS that explicitly account for decision confidence. This holds, e.g., for the two-way interaction between confidence and item values onto response time and choice (De Martino et al., 2012). Others may be more specific to oMCD (and related ideal control policies), e.g., the inversion of the value/confidence relationship for value-consistent and value-inconsistent choices. In any case, these non trivial second-order statistics are the hallmark of online/reactive control policies. In this context, what oMCD offers is a way to predict how these relationships should change, would effort costs and/or efficacies be experimentally manipulated.

We hope that this is what you had in mind.

Minor:

In figures, e.g. figure 6 where relationships between variables such as confidence and rt are shown, maybe put those on the axis labels in addition to or instead of the formulas. That's hard to read otherwise.

This relates to a similar point by reviewer #2: we have redone all the Figures, in the aim of improving the readability of our Results. In particular, we have now replaced mathematical formulas in axis labels with textual information...

Reviewer #2 (Remarks to the Author):

Summary

The manuscript presents an expanded version of a model for how people decide how much effort to invest in decisions. The focus is on value-based decisions, but as noted in the manuscript, the framework can be directly transferred to other kinds of decision. The extension is to consider a model that can reevaluate mid-trial how much time to spend deliberating (online MCD), as opposed to simply deciding before a trial begins, how much time to spend (prospective MCD). This seems like an important advance over the existing model. A very nice feature of the new model is that it is approximately valid for a range of specific decision-making mechanisms, as elaborated in the manuscript. I found these unifying derivations very innovative, and I find it an exciting line of work because it could support the parsimonious explanation of psychological phenomena in a range of tasks with a single model.

Overall, I think the manuscript offers a valuable contribution to the literature on balancing benefits and resource costs in decision-making, but I had some concerns which I think first need to be addressed. I list bigger points first and then smaller points.

Thank you for your positive summary!

Cognitive resources

I am not convinced that the framing of the paper accurately describes the manuscript's contribution. The stated aim of the manuscript is to explore control and specifically resource allocation. However, in the new online version of MCD, "resource" takes on a very limited definition of being equal to the time spent deliberating. The question for the observer is no longer how much resources to allocate, but how much time to spend deliberating. In the discussion the authors acknowledge that it is theoretically possible that observers spend more or less resources per unit time (through the parameter κ), but this possibility is not explored in the manuscript. Therefore, I am not sure that it is fair to say that the oMCD model is a model of the deployment of cognitive resources, when the only "resource" that observers have control over is decision time. I wonder whether a more accurate framing would be to present the model as a model of deliberation time.

This is a fair point. The way we think of this is that, in general, resource allocation is a threefold issue. In brief, one has to specify which resource to invest, with what intensity, and for how long. In this view, the current work simply focuses on the latter issue, and effectively ignores the other two. We have now added the following paragraph in the Discussion section to acknowledge this limitation (and we have removed the paragraph about κ that you are mentioning):

To begin with, recall that we have framed oMCD as a solution to a resource allocation problem. More precisely, we think of decision deliberation as involving the investment of costly cognitive resources, which are necessary to process decision-relevant information. The outcome of such resource allocation is to override default behavioral responses, which would otherwise be triggered by automatic (e.g., reflexive, habitual or intuitive) brain processes. Under this view, the brain faces the problem of adjusting *the amount* of resources to invest, which we equate with the issue of effort regulation. This perspective is not novel: the notion of mental effort was central to the early definition of automatic versus controlled processing, with the former described as quick and effortless, and the latter as slower and effortful (Schneider & Shiffrin, 1977). Since controlled processes are slow, it is reasonable to assume that the brain may regulate effort simply by adjusting its duration. This is the premise of our computational framework, which relies on the theory of optimal stopping (Shiryaev, 2007). However, effort actually unfolds along two dimensions: duration and intensity. This means that, in principle, both decision speed and confidence may be increased at the cost of increasing effort intensity. Accordingly, investing cognitive control is known to speed up responses in the context of, e.g., behavioral conflict tasks (Lin et al., 2022; Otto et al., 2022). This raises the question: what determines the brain's policy for trading effort intensity against effort duration? A possibility is that this depends upon the nature of the cognitive resource that is required for processing decision-relevant information. The issues of *how* to control resource investment and *which* resource to invest are thus intertwined (Shenhav et al., 2013). For example, one may think of resources as being composed of cognitive modules, such as working memory or attention, whose neurobiological underpinnings may induce distinct costs and/or limitations on effort intensity and duration (Grujic et al., 2022; Kool et al., 2017; Silvestrini et al., 2023). More generally, the effort intensity/duration tradeoff may be eventually determined by the neurobiological constraints that are imposed on the neural architecture that operates the processing of decision-relevant information (Musslick et al., 2015; Petri et al., 2017). For example, value-based decision making may require the active maintenance of multiple value representations that tend to interfere with each other, e.g., because they involve the same neural population within the orbitofrontal cortex (Pessiglione & Daunizeau, 2021b). In this case, cognitive control may alter the OFC neural code with the aim of temporarily dampening these interferences. In principle, the associated neural mechanism may operate based on simple confidence monitoring (which would proxy value conflict signals), without knowledge of the intricate architecture of value coding in the OFC. We will test these ideas using artificial neural network models of MCD in forthcoming publications.

Benefit term

The models all concern how observers trade cost for benefit. The models assume that the “benefit” that observers are aiming to maximise is confidence. The motivation behind this choice is not clear to me because, in the context of value-based decision making, there is a very clear and face-valid way to operationalise “benefit”. Specifically, “benefit” can simply be operationalised as obtained reward or obtained value. Probably because it is so straightforward, maximising reward/value has been the standard goal in previous work (e.g. Tajima et al., 2019a). My question is, what is the motivation for departing from this face-valid and standard measure of “benefit”?

In short, controlling decisions based upon a confidence gain is a more general strategy. For example, setting the benefit of resource allocation to the value of the chosen option does not make sense for evidence-based (e.g., perceptual) decisions. In contrast, setting the benefit of resource allocation to confidence makes sense for value-based decisions, and the ensuing control policy closely approximates the policy that maximizes the value of the chosen option. In other terms, oMCD is robust to this kind of violations of its working assumptions.

Having said this, we agree that we had not, until now, afforded evidence for the latter claim. In the revised manuscript, we compare oMCD with the optimal stopping policy that ensues from maximizing the value of the chosen option (so-called “*max(value)*” policy). We both show why (novel Appendix 3) and how (novel Results section 3) these two policies are almost equivalent.

In addition, we acknowledge that we did not properly motivate oMCD’s working assumption. Thus, we have now inserted the following paragraph in the Introduction section:

The working assumption here is that decision confidence serves as the main benefit term of the resource allocation problem (Lee et al., 2023; Yeung & Summerfield, 2012), hence the “metacognitive” nature of decision control. On the one hand, this formalizes the regulating role of confidence in decision making, which has recently been empirically demonstrated in the context of perceptual evidence accumulation (Balsdon et al., 2020, 2021). On the other hand, this apparently contrasts with standard treatments of value-based decision making, which insists on equating the benefit of value-based decisions with the value of the chosen option (De Martino & Cortese, 2022; Rangel et al., 2008a; Tajima et al., 2016a). This notion is a priori appealing, because the purpose of investing resources into decisions is reducible to approaching reward and/or avoiding losses/punishments. Nevertheless, the benefit of such resource investments may be detached from the subjective evaluation of alternative options (Smith & Krajbich, 2021). This is partly because the brain attaches subjective value to acquiring information about future rewards. In fact, this holds even when this information cannot be used to influence decision outcomes (Bennett et al., 2016; Bromberg-Martin et al., 2024; Jezzini et al., 2021). Recall that, in Marr’s sense, any type of decision induces the same computational problem, i.e. the comparison of alternative options. In this view, evidence-based and value-based decisions simply differ w.r.t. to the underlying comparison criterion: the former relies on truthfulness judgments while the latter involves idiosyncratic preferences (Summerfield & Tsetsos, 2012). Hence, in both cases, the benefit of allocating resources to decisions is to raise the chance of identifying the best option, i.e. confidence. In other words, if resource allocation aims at comparing alternative options, then decision confidence can be viewed as a probe for goal achievement. This is essentially a simplifying assumption, in the sense that it enables a unique computational architecture to control resource allocations, irrespective of the nature of the underlying decision-relevant computations.

Nature of the approximation

The manuscript nicely shows how different decision-making scenarios can be approximated under a more general model. I found these fascinating and very innovative results. In the manuscript it is argued that observers use this approximation because the relevant “controller” does not have knowledge of decision-making processes. The assumption that the controlling process has no knowledge of decision-processes is tenable, but it strikes me as a very strong assumption that deserves at least some justification or motivation. In particular, the idea that the controller has limited knowledge of the mechanisms of decision-making seems in tension with the common assumption that observers can use normative decision strategies by learning the statistical properties of decision-making tasks: If observers can learn the statistics of external stimuli, it would seem sensible to assume that they could also learn the statistics of their own decision-making mechanisms.

We agree that this is possible, provided the brain is equipped with decision-specific monitoring and control systems. But another alternative is that the brain relies on a unique but flexible metacognitive system, whose computational architecture is somehow recycled for addressing the needs of specific decision contexts. In our view, this issue is related to your previous comment. This is because we see oMCD’s reliance on the monitoring of confidence as a simplifying feature of the computational architecture of decision control (which shortcuts the underlying value computations). Accordingly, we have now inserted the following paragraph in the Discussion section of the revised manuscript:

One of the main assumptions behind MCD is that mental effort investment is regulated by a unique controller that operates under agnostic assumptions about the inner workings of the underlying decision system. This constraint somehow culminates in the simplicity of oMCD’s control architecture, which reduces to a monitoring of decision confidence. In this context, we have shown that the optimal stopping policies of distinct decision processes (*Bayesian value denoising* or *progressive attribute integration*) can be approximated using a simple calibration of effort efficacy parameters. We have also highlighted the ensuing properties of oMCD : when coupled with these different underlying decision systems, oMCD reproduces most established empirical results in the field of value-based decision-making. In addition, we have shown that oMCD is robust to alternative settings of the resource allocation problem. In particular, decision confidence seems to be a reasonable proxy for the value of the chosen option, which is the standard candidate titration for the benefit of value-value based decisions (Tajima et al., 2016b, 2019b). Taken together, these results suggest that the architecture of oMCD control, which relies on the internal monitoring of decision confidence, may generalize to most kinds of decision processes. Preliminary investigations show that this holds for yet another important kind of value-based decisions, whereby value computation is the output of a forward planning process on a decision tree (Consul et al., 2022; Sezener, 2018). Arguably, this also holds for perceptual or evidence-based decisions. In this context, decision confidence can be defined - somewhat more straightforwardly - as the subjective probability of being correct (Pouget et al., 2016). As long as effort efficacy parameters can be simply identified, the MCD architecture will provide an accurate approximation to the optimal resource allocation policy. This is trivial when perceptual detection or discrimination processes can be described as some form of *Bayesian denoising* of some perceptual variable of interest (Daunizeau et al., 2010; Drugowitsch et al., 2012). This would also hold for perceptual categorization processes, which may rather resemble *attribute integration* scenarios (Summerfield & Tsetsos, 2012). In fact, oMCD’s potential generalizability derives from its agnostic stance regarding the nature of information processing that takes place in the underlying decision system. This is also why oMCD can in principle be extended to describe the metacognitive control of other kinds of cognitive processes (e.g., reasoning or memory encoding/retrieval). In this context, an interesting avenue of investigation would be to consider the impact of metacognitive adaptation on the generalization of control policies across cognitive domains. Note that, because we assume MCD’s control architecture to be invariant across contexts, it requires a systematic calibration (in terms of, e.g., effort costs and/or efficacies) to guaranty the quasi-optimality of resource allocation. As we highlighted before, we expect such calibration to converge very quickly (e.g., over a few training trials). This is because effort efficacies can be learned from within-trial

confidence dynamics. Nevertheless, whether this specific kind of metacognitive adaptation is sufficient to recycle and adjust MCD's control architecture to novel cognitive domains, as well as how it shapes cross-domain metacognitive learning effects, is virtually unknown and would require specific empirical tests.

Testing / justifying approximations

Substantial approximations were made in the derivations of oMCD for the "ideal observer case" and "attribute-integration case". In particular I am thinking about the approximations on Ln 427 and Ln 489. However, as I understand the manuscript, the effect of these approximations was not directly studied or tested. For example, we do not know how much worse oMCD performs than the optimal solution for the two cases.

The effect of some of these approximations could be substantial. Consider the approximation on Ln 426. I assume that the range of admissible times is very big, i.e. the final admissible time, T , is very large (to ensure that this artificial deadline has little practical effect in the model). In this case, won't the average in Ln 427 be dominated by very large values of t , which are hardly ever realised? I.e. the average is over all admissible times, but most admissible times are never actually reached, with decisions usually happening much earlier. In this case, isn't the average quite a poor approximation? More generally, I wondered whether the authors had explored the effect of this approximation.

This is a fair point. In fact, our claim on oMCD's generalization ability would be vacuous if we could not show that it provides an accurate approximation to the ideal control policy (i.e. the control policy that relies on perfect knowledge of the underlying value computations). This is the reason why we have decided to identify the ideal control policy for both decision processes. This theoretical work, as well as the ensuing numerical validations, turned out to be quite time consuming, which is why this manuscript revision was delayed. Nevertheless, we thank you again for raising this point (and related issues) because we think that this motivated an effort that was worth investing in.

The mathematical derivations of the ideal policies can be found in Appendix 1 and 2, and the results of the ensuing analyses are reported in the first section of the revised Results section. In brief, we demonstrate why and how oMCD provides an accurate approximation to ideal stopping policies. We thus conclude that:

[...] the MCD architecture operates a quasi-optimal decision control that generalizes across decision processes without requiring detailed knowledge about underlying value computations.

Making assumptions clear

Sometimes assumptions were not clearly explained for the reader. Assumptions are of course crucial to any derivation, but it should be clear to the reader, what assumptions were made, and the ways in which they limit the scope of the derivations. For example, the text states (Ln 245) "Without loss of generality, we assume that there is a linear relationship between deliberation time and resource investment, i.e.: $z = kt$ ". But this is a strong loss of generality: The relationship could have been piecewise-linear or could have been any non-linear form. All these possibilities are being ruled out. To be clear, I am not asking the authors to reconsider which assumptions they made, just to make small changes to how they are presented.

On this, we slightly disagree with you. This is because these kinds of nonlinear effects would be almost redundant with the range of nonlinear effort costs that can be spanned by oMCD's current parametrization. Having said this, we acknowledge that this may be perceived as a contentious issue, and we have thus removed claims of "no loss of generality" in the revised manuscript. Note that the sentence that you refer to now reads:

Let t be the current time within a decision. We assume that there is a linear relationship between deliberation time and resource investment, i.e.: $z = \kappa t$, where κ is the amount of resources that is spent per unit of time.

Other examples:

- Ln 149 – 151 is another place where the text states "without loss of generality", but an assumption is made that reduces generality.

This was referring to the normal parametric form of probabilistic value representations. We have removed this claim. The corresponding sentence now reads:

In what follows, the probabilistic representation of option value takes the form of Gaussian probability density functions $p(V_i) = N(\mu_i, \sigma_i)$ where μ_i and σ_i are the mode and the variance of the probabilistic value representation, respectively (and i indexes alternative options in the choice set).

- Ln 176: "this implies that the precision of a given probabilistic value representation necessarily increases in proportion to the amount of allocated resources". What is necessary about this? Isn't in the case that various relationships would be possible (quadratic, generic monotonic, e.c.t.). This is a sensible assumption, but I don't see how it is "necessary".

You are right: the increase in precision is not necessarily linear. In fact, we simply meant that the precision needs to increase with the amount of resources. We have now modified this sentence as follows:

The former impact (i) derives from assuming that the amount of information that will be processed increases with the amount of allocated resources. This implies that the precision $1/\sigma_i(z)$ of a given probabilistic value representation necessarily increases with the amount of allocated resources [...]

In contrast, I felt the assumptions behind equation 4 (Ln 190 – 200) were introduced in a very clear, well-motivated manner.

Thank you!

Detailed points

- It could improve readability if the subplots were labelled A, B, C... and these referred to in the text, when a specific subplot in a figure is relevant.

We have now modified all the Figures accordingly.

- Ln 87 "kind of problems"  "kind of problem"? Thank you, corrected!

- Ln 96 “Interestingly, authors”  “Interestingly, the authors”? Corrected!

- Ln 98 Could it not be the case that the optimal threshold decreases very slowly, and hence a detectable decrease in confidence with time in empirical data is not necessarily expected?

Good point. But we believe this does not really modify the essence of the prediction of this work.

- Ln 100: There is also uncertainty regarding decision difficulty in Tajima et al., (2019a), isn't there? As I remember it, the strength of the value signal can vary, hence variability in decision difficulty. In any case, my understanding is that uncertainty regarding decision difficulty often generates optimal decision boundaries that decrease with time (E.g. Fig. 3 in Malhotra et al., 2018), which seems inconsistent with the argument being put forward here.

This is a fair comment. We have removed the corresponding sentence because it was not central to our argument anyway.

- Ln 106 “confidence reports as already been”  “confidence reports has already been”. Please check the text throughout for typographical errors. Thank you: corrected!

- Ln 128. Is the use of the word “disclaimer” necessary? It sounds very legal. Yes, we changed this.

- Ln 315. From this line it is clear that “a” and “pi” should be coded in the same way, but I believe the coding is inconsistent. Ln 269 says $a=0$ means the system stops. From Ln 281 I understand that $\pi=0$ means the system continues. Please correct me if I am wrong.

Thank you for spotting this typo: we have corrected it (and checked that this convention was satisfied throughout the remainder of the manuscript).

- Ln 309 / Equation 14. (A) I understood from above that $a=1$ means wait, but in the equation here (as I understand it) the $a=1$ case gives the reward associated with stopping. (B) From Equation 9 we have that $Q(\dots)=0$ if $a=1$. Hence, shouldn't at least one of the cases in Equation 14 be 0 (specifically the case associated with waiting). Same thing!

- Ln 382. I find “ideal observer case” a very confusing name for this case because, whatever information the observer receives, there will be some optimal way to process that information. For example, if the observer only receives information on how two options differ on specific attributes, there will still be an optimal way to use this attribute information that could then be described as an “ideal observer”. This case currently labelled “ideal observer” includes very specific assumptions (eg. Ln 389) which don't have any necessary connection to an ideal observer, but are rather substantive assumptions about decision-making mechanisms, and what information is received by them. Perhaps a better name would be something like “Gaussian noise case”.

We agree and have renamed this decision process the “Bayesian value denoising” case.

- Ln 586 – 587. What is the reason for excluding these no deliberation / no resource investment cases? They are still legitimate cases aren't they, and the model's strategy of no deliberation possibly normative?

You are right. We have reperformed this simulation series, without discarding the “no resource investment” trials.

- Ln 700. “decreases”  “increases”? Thank you: corrected!

- Ln 844: An approximately optimal solution, or an optimal solution? This paragraph of the Discussion has been entirely rewritten (see our responses above).

We thank you for your constructive criticisms, which have enabled us to significantly improve this manuscript!

Reviewer #3 (Remarks to the Author):

This is a review of the functionality of the code as per the invitation instructions: “to verify that the code included in this study is functional, that it is described in sufficient detail, and that it reproduces the results reported in the study”.

The supplementary materials for this manuscript contains the details to download code for this paper, and the toolbox used in that code, and details of running some demo scripts. The equations underlying the code are described appropriately in the manuscript, along with the steps taken in each simulation.

However, I cannot find the data/code availability statement in the main manuscript (as the Editorial Policy Checklist states is done), nor any reference to the software/language/toolboxes used for the simulations. This information is given in the reporting summary documents, but should be stated in the main manuscript also.

I could not reproduce all of the figure panels, as two scripts appeared to be missing from either the VBA toolbox or the oMCD code (VBA_plotxy.m and VBA_logisticReg.m, see error messages below). I could not find any reference to these scripts online at all.

Aside from those errors, I could reproduce all other figure panels, apart from Figure 1 which does not appear to have any code given in the readme file.

The download and installation works fine (though the VBA toolbox download they offer requires name & email in order to download from the authors website, which should not be required for blinded peer review, so perhaps this journal’s code requirements should not allow this in the future).

I have detailed the steps I took, and all warnings/errors I encountered below. I have run each demo to produce the figures in the paper, using Matlab R2021b on Windows 10.

Some scripts took far longer than the stated 1 minute to run, due to the warnings produced by using a deprecated function in the VBA toolbox (see warnings below). Manually turning off these warnings greatly sped up the scripts to well below 1 minute (e.g. demo_MCDs.m took way over 20 minutes to run, which became 32 seconds after commenting those warnings out). It seems like the functions ‘sig.m’ and ‘vec.m’ are wrapper functions to call ‘VBA_sigmoid.m’ and ‘VBA_vec.m’ and print the warning, so they should probably be replaced with those newer scripts throughout the oMCD code.

Warning: *** The function `sig` is now deprecated. Please see
`VBA_sigmoid` for an alternative.

> In sig (line 4)

In getPc>myEabs2 (line 32)

In getPc (line 19)

In demo_MCDs (line 71)

Warning: *** The function `vec` is now deprecated and has been renamed `VBA_vec`.

> In vec (line 4)

In get_wMDPO (line 23)

In demo_MCDs (line 53)

I ran each demo in the readme:

1. No code is presented to replicate Figure 1
2. demo_Econf.m runs fine (these figures are not in the paper)
3. demo_onlineMCD0.m reproduces figure 2, but then gave an error during plotting Figure 3d (panels a-c are reproduced though), as it appears the function 'VBA_plotxy' is not included in either the toolbox or the code (see error message below). Figure 4 is reproduced if the cell after this error is run.

Unrecognized function or variable 'VBA_plotxy'.

Error in demo_onlineMCD0 (line 168)

```
out = VBA_plotxy(t_omcd,P_omcd,10,ha,0);
```

4. demo_MCDs.m ran fine and reproduced figure 5 (plus an additional panel which does not appear in the manuscript)
5. demo_oMCD_difficulty.m reproduced Figure 6a-b but gave the 'VBA_plotxy' error (see above point #3 above, occurred on line 114 in the current script), so could not make panel C. Panel d was made if the code after that error was run (although the legend did not contain all the different effort level labels, due to that error halting the code). Figure 7 can be reproduced by changing all the parameters at the top to 2 (apart from panel C due to this missing script).
6. demo_oMCD_thresholds.m reproduces Figure 8 (as a side note, I would recommend moving the legends in Figure 8a+b to not cover the magenta line).
7. demo_oMCD_Vupdates.m again has the VBA_plotxy error (line 90), as well as a missing function VBA_logisticReg.m (line 106), so I am only able to reproduce panels a of Figures 9 and 10

Thank you very much for this thorough review of our code. We have now modified the attached code to include the two missing functions. And we have turned the warnings off to improve code execution speed.

Note that our revised work includes extra code, which we document in the same manner as before.

13th Feb 24

Dear Jean,

Your manuscript titled "The online metacognitive control of decisions" has now been seen by our reviewers, whose comments appear below. In light of their advice I am delighted to say that we are happy, in principle, to publish a suitably revised version in Communications Psychology under the open access CC BY license (Creative Commons Attribution v4.0 International License).

We therefore invite you to revise your paper one last time to address the remaining concerns of our reviewers and a list of editorial requests. At the same time we ask that you edit your manuscript to comply with our format requirements and to maximise the accessibility and therefore the impact of your work.

EDITORIAL REQUESTS:

SUBMISSION INFORMATION:

OPEN ACCESS:

Communications Psychology is a fully open access journal. Articles are made freely accessible on publication under a CC BY license (Creative Commons Attribution 4.0 International License). This license allows maximum dissemination and re-use of open access materials and is preferred by many research funding bodies.

For further information about article processing charges, open access funding, and advice and support from Nature Research, please visit <https://www.nature.com/commspsychol/article-processing-charges>

At acceptance, you will be provided with instructions for completing this CC BY license on behalf of all authors. This grants us the necessary permissions to publish your paper. Additionally, you will be asked to declare that all required third party permissions have been obtained, and to provide billing information in order to pay the article-processing charge (APC).

* TRANSPARENT PEER REVIEW: Communications Psychology uses a transparent peer review system. On author request, confidential information and data can be removed from the published reviewer

reports and rebuttal letters prior to publication. If you are concerned about the release of confidential data, please let us know specifically what information you would like to have removed. Please note that we cannot incorporate redactions for any other reasons.

* CODE AVAILABILITY: All Communications Psychology manuscripts must include a section titled "Code Availability" at the end of the methods section. We require that the custom analysis code supporting your conclusions is made available in a publicly accessible repository at this stage; please choose a repository that generates a digital object identifier (DOI) for the code; the link to the repository and the DOI must be included in the Code Availability statement. Publication as Supplementary Information will not suffice.

* DATA AVAILABILITY:

[link redacted]

Best regards,

Marike

Marike Schiffer, PhD
Chief Editor
Communications Psychology

REVIEWERS' COMMENTS:

Reviewer #1 (Remarks to the Author):

The authors have addressed my concern and I find the manuscript much improved. I believe the the figure legibility could still be improved a little bit (e.g. for Figure 6, it would be helpful to give a pointer to what beta is for people who read manuscripts looking through figures first [or only]), but these are minor. Overall, I believe this manuscript makes for a nice contribution to the literature.

Reviewer #2 (Remarks to the Author):

I already felt that the original manuscript made an original and valuable contribution to the field. The extensive revisions made by the authors have further substantially strengthened the paper. The revisions are very comprehensive, and I am grateful to the authors for carefully addressing all the points I raised. I found it particularly interesting to read the author's reasoning for placing decision confidence at the centre of the model. It is a very nice point that this makes the model more generally applicable. The results comparing the oMCD to the optimal models for specific situations are also very enlightening, and it is impressive how well the oMCD performs in these comparisons. Apart from the minor points that I raise below, I would recommend the manuscript for acceptance.

Ln 50. "Recent theoretical 50 neuroscience work propose to view" -> "proposes to view"?

Ln 116 "The net benefit of decision time is the estimated reward rate". I found this sentence hard to understand, specifically I think I was confused by the "net benefit" term. By "net benefit" do you mean the cost per unit time of deliberating?

Ln 128 "shortcut summary statistics" -> "shortcut summary statistic"?

Ln 222. Equation was not rendered correctly for me. There was a box on the second line of the equation. Similarly on Ln 249.

Ln 415. "In this example, this relationship will be mostly negative, i.e. reported confidence levels will decrease when response times increase. This is despite the fact that expected confidence always increases with decision time". I understand what you mean here because I am familiar with the literature, but for readers less familiar with the literature this could be a very confusing sentence: It sounds like confidence both increases and decreases with response time at the same time.

Section 2 and 3 of the results. Sets of parameters were repeatedly randomly drawn for the simulations, but I did not see it reported anywhere, what distributions these parameters were drawn from. (If this is clear in the code published alongside the paper, perhaps this is not necessary to include.)

Ln 466 and 468. No derivation is provided for Eq. 20 or 21. Is there are relevant citation for these derivations that can be given?

Ln 822. "Error! Reference source not found."

Ln 1010 Typo: "the optimal sopping policy"

Ln 1025. I am wondering whether the types of lines above the delta symbols have got mixed up. In Ln 1025 the expectation of the accumulated instantaneous (i.e., total) perturbations is taken, but shouldn't this be the expectation of a single instantaneous perturbation? Only in this case is the expectation 0, right? (Relatedly, is the symbol on Ln 483 correct? This is the symbol for the total perturbation, not instantaneous perturbation.)

Ln 1028. Similarly, the symbol for the accumulated instantaneous (total) perturbations is shown in the expectation (i.e. delta with a tilde), but the text says that this equation gives the variance of an individual instantaneous perturbation (which is depicted by a delta with a different line over the top of it).

Reviewer #3 (Remarks to the Author):

This is a review of the functionality of the code as per the invitation instructions: “to verify that the code included in this study is functional, that it is described in sufficient detail, and that it reproduces the results reported in the study”.

The code all runs fine now that the previously missing functions are included and that warning that was slowing it down was removed. The only issues I found were minor things in the readme_oMCD.doc (and the supplementary materials I received which had a copy of this readme), not in the code itself, which I have detailed below. Additionally, the demo_valueComputations0.m file took several minutes to run, due to the monte-carlo simulation loop on line 157, rather than the <1 stated in the readme.

I was able to reproduce all the figures.

I have downloaded the updated VBA toolbox (the ‘personal version’ from <https://sites.google.com/site/jeandaunizeauswebsite/code>). Thank you for updating the ‘sig.m’ file to speed up the scripts, and for including those previously missing functions.

Minor issues:

1. demo_MDP0.m creates Figure 1 (but this file is misnamed as ‘demo_MDD0.m’ in the readme and supplementary information I received (which seems to be a copy of this readme))
2. demo_oMCD_Vupdates.m reproduces Figures 10 & 11, not figures 7 & 8 as the readme states
3. demo_value Computations0.m – typo (space) in file name in readme + supplemental. Otherwise reproduces figures fine

Title : The online metacognitive control of decisions

Journal: Communications Psychology

Manuscript id: COMMSPSYCHOL-23-0088-T

Reviewer #1 (Remarks to the Author):

The authors have addressed my concern and I find the manuscript much improved. I believe the the figure legibility could still be improved a little bit (e.g. for Figure 6, it would be helpful to give a pointer to what beta is for people who read manuscripts looking through figures first [or only]), but these are minor. Overall, I believe this manuscript makes for a nice contribution to the literature.

Thank you for your positive appreciation of our work!

Re: Figure 6: we have slightly modified the caption to clarify things.

Reviewer #2 (Remarks to the Author):

I already felt that the original manuscript made an original and valuable contribution to the field. The extensive revisions made by the authors have further substantially strengthened the paper. The revisions are very comprehensive, and I am grateful to the authors for carefully addressing all the points I raised. I found it particularly interesting to read the author's reasoning for placing decision confidence at the centre of the model. It is a very nice point that this makes the model more generally applicable. The results comparing the oMCD to the optimal models for specific situations are also very enlightening, and it is impressive how well the oMCD performs in these comparisons. Apart from the minor points that I raise below, I would recommend the manuscript for acceptance.

Thank you for your positive appreciation of our work!

Ln 50. "Recent theoretical 50 neuroscience work propose to view" -> "proposes to view"? **Thank you: corrected!**

Ln 116 "The net benefit of decision time is the estimated reward rate". I found this sentence hard to understand, specifically I think I was confused by the "net benefit" term. By "net benefit" do you mean the cost per unit time of deliberating? **By "net benefit", we mean the reward discounted by deliberation time. We changed this sentence to "[...] (iii) the net benefit of decisions (i.e. the benefit discounted by decision time) is the estimated reward rate".**

Ln 128 "shortcut summary statistics" -> "shortcut summary statistic"? **Thank you: corrected!**

Ln 222. Equation was not rendered correctly for me. There was a box on the second line of the equation. Similarly on Ln 249. This seems to be a typo induced by MathType (the software we use for Equations editing). The sign that should appear is “ \sim ”, which means “is sampled from”.

Ln 415. “In this example, this relationship will be mostly negative, i.e. reported confidence levels will decrease when response times increase. This is despite the fact that expected confidence always increases with decision time”. I understand what you mean here because I am familiar with the literature, but for readers less familiar with the literature this could be a very confusing sentence: It sounds like confidence both increases and decreases with response time at the same time. Good point: we acknowledge that this paragraph may induce confusion. We changed it as follows:

In this example, this relationship will be mostly negative, i.e. reported confidence levels tend to decrease when response times increase. This is despite the fact that average confidence $\bar{P}_c(t)$ always increases as decision time unfolds, as long as effort efficacy parameters are nonzero. In other words, the overt relationship between response times and reported confidence levels (across trials) may be qualitatively different from the covert temporal dynamics of confidence during decision deliberation.

Section 2 and 3 of the results. Sets of parameters were repeatedly randomly drawn for the simulations, but I did not see it reported anywhere, what distributions these parameters were drawn from. (If this is clear in the code published alongside the paper, perhaps this is not necessary to include.)

Sampling distributions of model parameters are of course specified in the code.

Ln 466 and 468. No derivation is provided for Eq. 20 or 21. Is there are relevant citation for these derivations that can be given?

Equations 20 and 21 trivially derive from standard derivations from Bayesian inference under conjugate Gaussian models. We now provide a citation of our own work that uses a similar derivation:

Daunizeau, Jean, Hanneke E. M. den Ouden, Mathias Pessiglione, Stefan J. Kiebel, Klaas E. Stephan, et Karl J. Friston. « Observing the observer (I): meta-bayesian models of learning and decision-making. » *PloS one* 5, n° 12 (2010): e15554.

Ln 822. “Error! Reference source not found.” Thank you: this was an automatic Equation referencing error (corrected)!

Ln 1010 Typo: “the optimal sopping policy” Thank you: corrected!

Ln 1025. I am wondering whether the types of lines above the delta symbols have got mixed up. In Ln 1025 the expectation of the accumulated instantaneous (i.e., total) perturbations is taken, but

shouldn't this be the expectation of a single instantaneous perturbation? Only in this case is the expectation 0, right? Well spotted: we have corrected this typo!

Relatedly, is the symbol on Ln 483 correct? This is the symbol for the total perturbation, not instantaneous perturbation. Here, the notation is correct.

Ln 1028. Similarly, the symbol for the accumulated instantaneous (total) perturbations is shown in the expectation (i.e. delta with a tilde), but the text says that this equation gives the variance of an individual instantaneous perturbation (which is depicted by a delta with a different line over the top of it). Same error as above, which we have now corrected (thank you)!

Reviewer #3 (Remarks to the Author):

This is a review of the functionality of the code as per the invitation instructions: "to verify that the code included in this study is functional, that it is described in sufficient detail, and that it reproduces the results reported in the study".

The code all runs fine now that the previously missing functions are included and that warning that was slowing it down was removed. The only issues I found were minor things in the readme_oMCD.doc (and the supplementary materials I received which had a copy of this readme), not in the code itself, which I have detailed below. Additionally, the demo_valueComputations0.m file took several minutes to run, due to the monte-carlo simulation loop on line 157, rather than the <1 stated in the readme.

I was able to reproduce all the figures.

I have downloaded the updated VBA toolbox (the 'personal version' from <https://sites.google.com/site/jeandaunizeauswebsite/code>). Thank you for updating the 'sig.m' file to speed up the scripts, and for including those previously missing functions.

Minor issues:

1. demo_MDPO.m creates Figure 1 (but this file is misnamed as 'demo_MDD0.m' in the readme and supplementary information I received (which seems to be a copy of this readme))
2. demo_oMCD_Vupdates.m reproduces Figures 10 & 11, not figures 7 & 8 as the readme states
3. demo_value Computations0.m – typo (space) in file name in readme + supplemental. Otherwise reproduces figures fine

Thank you very much for this thorough review of our code. We have now corrected the typos in the revised readme file.